# Exploring and Exploiting Model Uncertainty in Bayesian Optimization

**Zishi Zhang**[1]*, **Tao Ren**[1]*, **Yijie Peng**[1,2]†
[1] Guanghua School of Management, Peking University
[2] Xiangjiang Laboratory
{zishizhang,rtkenny}@stu.pku.edu.cn, pengyijie@pku.edu.cn

## Abstract

In this work, we consider the problem of Bayesian Optimization (BO) under *reward model uncertainty*—that is, when the underlying distribution type of the reward is unknown and potentially intractable to specify. This challenge is particularly evident in many modern applications, where the reward distribution is highly ill-behaved, often non-stationary, multi-modal, or heavy-tailed. In such settings, classical Gaussian Process (GP)-based BO methods often fail due to their strong modeling assumptions. To address this challenge, we propose a novel surrogate model, the infinity-Gaussian Process ($\infty$-GP), which represents a sequential spatial Dirichlet Process mixture with a GP baseline. The $\infty$-GP quantifies both value uncertainty and model uncertainty, enabling more flexible modeling of complex reward structures. Combined with Thompson Sampling, the $\infty$-GP facilitates principled exploration and exploitation in the distributional space of reward models. Theoretically, we prove that the $\infty$-GP surrogate model can approximate a broad class of reward distributions by effectively exploring the distribution space, achieving near-minimax-optimal posterior contraction rates. Empirically, our method outperforms state-of-the-art approaches in various challenging scenarios, including highly non-stationary and heavy-tailed reward settings where classical GP-based BO often fails.

## 1 Introduction

Bayesian Optimization (BO) [Frazier, 2018] is a powerful framework for optimizing expensive-to-evaluate black-box functions and has found broad applications in hyperparameter tuning for machine learning models [Snoek et al., 2012], robotics [Berkenkamp et al., 2023], biology [Amin et al., 2024], and reinforcement learning [Brochu et al., 2010]. BO typically proceeds by fitting a surrogate model to the observed reward collected so far, and then using an acquisition policy to guide the selection of the next location to query. The surrogate model plays a central role in both learning and exploration.

In almost all BO applications, Gaussian Processes (GPs) are used as the default surrogate model. This practice assumes that the deterministic objective function $\mu^*(x)$ and the noise $\epsilon^*(x)$ satisfy strong structural conditions, essentially requiring the true objective to "resemble" a GP. Specifically, $\mu^*(x)$ is often assumed to be a sample path drawn from GP [Russo and Van Roy, 2014, Wang et al., 2025] or lie in a reproducing kernel Hilbert space (RKHS) with known bounded norm [Srinivas et al., 2010, 2012, Chowdhury and Gopalan, 2017] (the connection between GP regression and RKHS regression is detailed in Kanagawa et al. [2018]), and $\epsilon^*(x)$ is assumed to be independent and (sub-)Gaussian, or even noiseless [Chen and Lam, 2023]. These assumptions are critical for theoretical guarantees,

---

* These authors contributed equally to this work.
† Corresponding author.

39th Conference on Neural Information Processing Systems (NeurIPS 2025).

but are rarely verified in practice. GP-based BO can fail when these assumptions are violated. For instance, GP surrogates struggle to model non-stationary rewards, which are commonly observed in applying BO in machine learning hyperparameter tuning [Snoek et al., 2014], and are inadequate for handling heavy-tailed noise, which frequently arises in financial and real-world BO tasks [Chowdhury and Gopalan, 2019, Cakmak et al., 2020]. Moreover, prompt optimization for large language models (LLMs) is another representative application [Sabbatella et al., 2024] where the reward model is of unknown and complex form. The reward typically involves multiple stages—such as combining with inputs, querying a black-box LLM, and interpreting outputs—each introducing variation and instability. The resulting reward landscape is highly unstable and non-stationary [Deng et al., 2022], as even slight variations in prompt formulation can lead to dramatically different outputs.

In this paper, we consider BO under *reward model uncertainty*—that is, the decision-maker is uncertain not only about the parameters of the reward distribution, but also about its underlying distributional type. While classical GP-based BO can effectively quantify *value uncertainty* under a fixed model, it fails to capture *model uncertainty*. Our goal is to develop a surrogate modeling and decision-making framework that remains robust under *reward model uncertainty*. To this end, we propose a novel surrogate model, the $\infty$-*Gaussian Process* ($\infty$-GP), which represents an infinite mixture of Gaussian processes implemented via a sequential spatial Dirichlet process prior with a GP baseline. When combined with Thompson Sampling, this model automatically facilitates a principled exploration–exploitation trade-off in the distributional space, and can effectively converge toward a wide range of reward models.

**Related Work.**    To address non-stationarity in BO, prior works have proposed various modifications, such as warping the input space of standard GP kernels [Snoek et al., 2014], replacing the surrogate model with neural networks [Li et al., 2023], or employing non-stationary kernels [Higdon et al., 2022, Seeger, 2004]. To handle heavy-tailed noise, a substantial body of work has been developed in the context of multi-armed bandits, i.e., problems with a finite decision space [Bubeck et al., 2013, Medina and Yang, 2016]. However, for BO with a continuous decision space, the available methods are much limited. One notable exception is the work of Chowdhury and Gopalan [2019], who propose a truncation-based modification to the GP-UCB algorithm to handle heavy-tailed noises. Misspecified BO [Wynne et al., 2021, Bogunovic and Krause, 2021] addresses limited forms of model mismatch (e.g., incorrect smoothness) within fixed model classes such as GPs or RKHS functions, rather than accounting for distributional uncertainty over the entire distributional space. Mixture-of-experts GP models [Rasmussen and Ghahramani, 2001, Meeds and Osindero, 2005] also construct infinite GP mixtures using Dirichlet processes. However, they rely on a global DP to partition the input space, assigning each input to a single GP expert. In contrast, our model employs a spatial DP prior, allowing each location to mix over infinitely many GP surfaces. This leads to a fundamentally different formulation that offers a richer, nonparametric representation of model uncertainty and enables convergence over a broad class of reward distributions. Bariletto and Ho [2024] also considers combining nonparametric Bayesian models, such as DP, with BO. However, their focus is on distributionally robust optimization, and the integration is carried out fundamentally differently from ours.

**Our Contributions**    The main contributions of this work are as follows:

- **A novel $\infty$-GP surrogate model.** We propose a new surrogate model for BO, the $\infty$-Gaussian Process ($\infty$-GP), which quantifies both *model uncertainty* and *value uncertainty*. Combined with Thompson Sampling, it enables principled and efficient exploration and exploitation in the distributional space of the reward.

- **Theoretical guarantees.** We establish posterior convergence guarantees for the $\infty$-GP model, showing that it can approximate a broad class of reward distributions at a near-optimal minimax rate.

- **Strong empirical performance.** We demonstrate the effectiveness of our method on both synthetic benchmarks and real-world tasks where standard GP-based BO fails—particularly in scenarios involving heavy-tailed noise and non-stationary reward structures.

## 2 Preliminaries

**Bayesian Optimization (BO).** We consider the problem of maximizing an expensive-to-evaluate black-box function $\mu^*(x)$ over a compact domain $\mathcal{X} \subset \mathbb{R}^d$. At each iteration, the decision maker selects a point $x_i \in \mathcal{X}$ and observes a noisy reward $y(x_i) = \mu^*(x_i) + \epsilon^*(x_i)$, where $\epsilon^*(x)$ denotes stochastic noise. Since querying the function is costly, the goal is to identify the maximizer of $\mu^*(x)$ using as few evaluations as possible. BO tackles this problem by maintaining a probabilistic model—called a *surrogate model*—over the unknown objective. This surrogate is updated after each evaluation and serves as a cheap proxy for the reward. Based on the surrogate, an *acquisition policy* is used to determine the next query location. The policy aims to balance *exploration* (sampling uncertain regions) and *exploitation* (focusing on areas with high predicted values), thereby efficiently searching for the global optimum. Common acquisition strategies include Expected Improvement (EI), Upper Confidence Bound (UCB), Knowledge Gradient (KG), and Thompson Sampling (TS), which draws a random sample from the posterior and selects the maximizer.

**Gaussian Process Surrogate Modeling and Kriging.** Standard BO methods adopt a Gaussian Process (GP) as the surrogate model. The observed reward is modeled as $y(x_i) = m(x_i) + \xi(x_i) + \epsilon_i$, $\epsilon_i \sim \mathcal{N}(0, \tau^2)$, where $m(x) = \beta^\top x$ denotes the deterministic mean trend, and $\xi \sim \mathcal{GP}(0, \sigma^2 \rho_\phi)$ is a zero-mean stationary GP over $\mathcal{X}$ with squared exponential kernel $\rho_\phi(x, x') = \exp\left(-\sum_{k=1}^d \phi_k (x^{(k)} - x'^{(k)})^2\right)$, $\phi = (\phi_1, \cdots, \phi_d)$, $x = (x^{(1)}, \cdots, x^{(d)})$. The combined term $m(x) + \xi(x)$ acts as a probabilistic surrogate for the true reward function $\mu^*(x)$. A key advantage of GP models lies in their closed-form posterior inference: given realized historical data $\xi(x_{1:n}) = \{\xi(x_1), \cdots, \xi(x_n)\}$ at evaluated locations $x_{1:n} = (x_1, \cdots, x_n)$, the posterior distribution of the entire function $\xi(\cdot)$ remains a GP. This is known as ***Kriging*** (or GP regression):

$$[\xi(\cdot) \mid \xi(x_{1:n})] \sim \mathcal{GP}(\mu_n(\cdot), \sigma_n^2(\cdot)), \tag{1}$$

$$\mu_n(\cdot) = \Sigma_0(\cdot, x_{1:n}) \Sigma_0^{-1}(x_{1:n}, x_{1:n}) \xi(x_{1:n}), \tag{2}$$

$$\sigma_n^2(\cdot) = \sigma^2 - \Sigma_0(\cdot, x_{1:n}) \Sigma_0^{-1}(x_{1:n}, x_{1:n}) \Sigma_0(x_{1:n}, \cdot). \tag{3}$$

Here, the kernel is $\Sigma_0(x, x') := \sigma^2 \rho_\phi(x, x')$. The matrix $\Sigma_0(x_{1:n}, x_{1:n}) \in \mathbb{R}^{n \times n}$ denotes the covariance matrix with entries $[\Sigma_0]_{ij} = \Sigma_0(x_i, x_j)$, and $\Sigma_0(x, x_{1:n}) \in \mathbb{R}^{1 \times n}$ is the row vector of covariances between $x$ and each observed input $x_i$.

**Value Uncertainty vs. Model Uncertainty** Another key strength of GP in BO is their ability to quantify uncertainty through the posterior variance. This enables principled exploration–exploitation (E&E) trade-offs, as in the UCB strategy, which directly uses the variance to guide exploration. However, the uncertainty quantified by a GP is inherently limited to *value uncertainty*—that is, uncertainty about the response value at a given location under a fixed reward distribution model. This overlooks a more fundamental source of uncertainty: whether the GP itself is an appropriate surrogate model for the underlying reward. We refer to this as *model uncertainty*, which captures the decision maker's uncertainty belief about the type or complexity of the true reward distribution. Consequently, standard GP-based BO requires strong assumptions for convergence, and GPs fail to capture complex reward models, such as those that are **non-stationary** or contaminated with **heavy-tailed noises**.

## 3 $\infty$-GP Surrogate Modeling

In this section, we develop a novel surrogate model called the $\infty$-Gaussian-Process ($\infty$-GP), formally referred to as the *sequential spatial Dirichlet Process mixture with a Gaussian Process baseline*. The observed reward is modeled as

$$y(x_i) = m(x_i) + \xi(x_i) + \epsilon_i, \ \epsilon_i \sim N(0, \tau^2), \quad \xi(x_i) \overset{\text{ind}}{\sim} G_{x_i}, \tag{4}$$

where the deterministic mean term $m(x_i)$ is the same as classic GP model and the stochastic process $\xi \triangleq \{\xi(x) \in \mathbb{R} : x \in \mathcal{X}\}$ follows distribution $\{G_x : x \in \mathcal{X}\}$. To capture **model uncertainty** and explore the space of reward distributions, we do not fix the distribution $\{G_x : x \in \mathcal{X}\}$ as a GP. Instead, we assume that **the distribution $\{G_x : x \in \mathcal{X}\}$ is itself random** and follows a sequential Spatial Dirichlet Process (SDP) prior, which places a prior over the space of distributions:

$$\{G_x : x \in \mathcal{X}\} \sim \text{SDP}(\nu G_0). \tag{5}$$

Therefore, each reward $y(x_i)$ is generated via the following hierarchical process: (i) the nature draws a stochastic process $\{G_x : x \in \mathcal{X}\}$ from the distribution space under a SDP prior. (ii) in the $i$-th iteration, after determining the location $x_i$ to evaluate, a sample $\xi(x_i)$ is generated from distribution $G_{x_i}$.

The SDP$(\nu G_0)$ is characterized by two key quantities: a scalar precision parameter $\nu > 0$, and a baseline distribution $G_0$. In this work, we specify a stationary Gaussian Process $\mathcal{GP}(0, \sigma^2 \rho_\phi(\cdot, \cdot))$ over $\mathcal{X}$ as the baseline distribution. Specifically, a distribution (or stochastic process) arising from SDP$(\nu G_0)$ is almost surely discrete and admits the representation

$$G_x = \sum_{l=1}^{\infty} w_l \delta_{\xi^{(l)}(x)}, \forall x \in \mathcal{X}, \tag{6}$$

where each surface $\xi^{(l)} \triangleq \{\xi^{(l)}(x) : x \in \mathcal{X}\}$ is a sample path independently drawn from the baseline $\mathcal{GP}(0, \sigma^2 \rho_\phi(\cdot, \cdot))$ and the weights $\{w_l\}_{l=1}^{\infty}$ admits the traditional "stick breaking" construction of Dirichlet process, i.e., $w_1 = V_1$, $w_l = V_l \prod_{r}^{l-1}(1 - V_r)$, $V_r \overset{iid}{\sim} Beta(1, \nu)$. Notably, our approach directly models the distribution of the observed reward $y(x)$, rather than $\mu^*(x)$. In this context, the additive noise term $\epsilon_i \sim N(0, \tau^2)$ in model (4) is not intended to impose a Gaussian assumption on the true observation noise. Instead, it is solely introduced for modeling purposes—to mix with the discrete-distributed $\xi(x)$ and produce a continuous reward distribution.

As defined in (6), each realized distribution $\{G_x : x \in \mathcal{X}\}$ from the SDP is a discrete measure over an infinite collection of surfaces $\{\xi^{(l)}\}_{l=1}^{\infty}$. At any location $x_i$, there is a positive probability that $\xi(x_i) = \xi^{(l)}(x_i)$ for some $l$, meaning that a draw $\xi(x_i) \sim G_{x_i}$ can be realized on any of the surfaces in the collection. To identify which surface $x_i$ lies on, we introduce a latent variable $z_{1:n} = \{z_i\}_{i=1}^{n}$, such that $z_i = l$ if $\xi(x_i) = \xi^{(l)}(x_i)$ and thus $\xi^{(z_i)}$ is the surface on which observation $x_i$ is realized. Note that for any surface $\xi^{(j)}$ on which the observation at $x_i$ is not realized, the corresponding value $\xi^{(j)}(x_i)$ remains latent and must also be inferred. As illustrated in Figure 1, solid dots (e.g., $\xi^{(2)}(x_5)$) indicate that the observation at $x_5$ is realized on surface $\xi^{(2)}$, while hollow dots (e.g., $\xi^{(1)}(x_3)$) indicate that the observation at $x_3$ is not realized on surface $\xi^{(1)}$. Since multiple observations may be realized on the same surface, the number of realized surfaces, denoted by $K_n$, typically grows slowly with $n$. Let $\boldsymbol{\xi}_{1:n} = (\xi^{(1)}, \ldots, \xi^{(K_n)})$ denote the collection of realized surfaces up to iteration $n$, and let $n_j$ denote the number of locations within $\{x_1, \cdots, x_n\}$ that are realized on surface $\xi^{(j)}$, i.e., $n_j = \#\{i : z_i = j\}, j = 1, \cdots, K_n$.

It is important to note that the SDP was initially introduced in the spatial statistics and geostatistics literature [Gelfand et al., 2005, Quintana et al., 2022]. Our $\infty$-GP model incorporates two key modifications to adapt SDP for the sequential nature of BO. First, instead of assuming a fixed set of locations, we allow observations to be collected sequentially at arbitrary locations. Second, unlike the original SDP, where observations across all locations in each replication are drawn from the same surface, our model permits each observation to be realized on a potentially different surface, enabling greater flexibility.

**Model Uncertainty Quantification** It is important to understand the two quantities that characterize the SDP: the baseline distribution $G_0$ and the concentration parameter $\nu$. For any measurable set $B$, if $G \sim$ SDP$(\nu G_0)$, then $\mathbb{E}[G(B)] = G_0(B)$, meaning that $G_0$ serves as the mean measure. The parameter $\nu$ can be interpreted as the "variance" in the distribution space, controlling the variability of $G$ around $G_0$: $\text{Var}(G(B)) = \frac{G_0(B)(1-G_0(B))}{1+\nu}$. In practice, updating the posterior of $\nu$ with data allows dynamic quantification of *model uncertainty*: **the higher the $\nu$, the more confident we are that the true reward distribution follows $G_0$**. In contrast, a small $\nu$ implies significant model uncertainty, in which case the model relies primarily on empirical distributions (detailed in the next section). Notably, although the baseline $G_0$ is a stationary GP, a sample path drawn from the $\infty$-GP can be **non-stationary** and exhibit different smoothness. This highlights the enhanced modeling flexibility of the $\infty$-GP surrogate.

**Full Bayesian Treatment of Hyperparameters** Following the seminal work of Snoek et al. [2012], we adopt a fully Bayesian approach for the hyperparameters $\Theta = \{\beta, \nu, \tau, \sigma^2, \phi\} = \{\Theta^{(1)}, \Theta^{(2)}\}$, where the first-layer parameters are $\Theta^{(1)} = \{\beta, \tau\}$ and the second-layer parameters are $\Theta^{(2)} =$

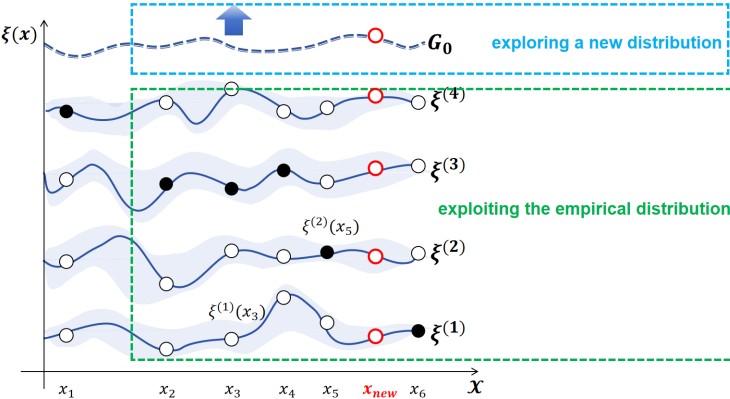

Figure 1: Illustration of the proposed $\infty$-GP model as a mixture over infinitely many GP surfaces. Each observation may be realized on any surface—solid dots indicate assignments to the corresponding surface, while hollow dots represent latent observations not realized on that surface. The red dot denotes a new input location $x_{\text{new}}$, which may either be assigned to an existing surface or initiate a new one drawn from the base measure.

$\{\nu, \sigma^2, \phi\}$. Instead of relying on frequentist estimators such as MLE, we place prior distributions on all components of $\Theta$ and update them via posterior inference. Details of the prior specifications are deferred to the Appendix.

## 4 $\infty$-GP Thompson Sampling: E&E in Distributional Space

Although the surrogate modeling in the previous section is performed on the noisy observed reward $y(x)$, the goal in BO is to maximize the expected reward $\mu^*(x) := \mathbb{E}[y(x)]$. The corresponding Thompson Sampling (TS) acquisition policy proceeds by drawing a sample from the posterior distribution of $\mu^*(x)$ given data $\mathcal{H}_n = \{(x_1, y(x_1)), \cdots, (x_n, y(x_n))\}$, and selecting its maximizer as the next query point, i.e.

$$x_{n+1} = \arg\max_{x \in \mathcal{X}} \hat{\mu}^*(x), \quad \hat{\mu}^* \sim p(\mu^* \mid \mathcal{H}_n). \tag{7}$$

In our hierarchical Bayesian model, at the $(n + 1)$-th step, for any unobserved location $x_{n+1} \in \mathcal{X}$, the posterior distribution $\mu^*(x_{n+1}) \mid \mathcal{H}_n$ is implicitly induced by the joint posterior over latent components $\Theta' := \{\xi^{(z_{n+1})}(x_{n+1}), \Theta^{(1)}\}$, where $\xi^{(z_{n+1})}$ denotes the surface on which $x_{n+1}$ is realized and $\Theta^{(1)} = \{\beta, \tau^2\}$. Specifically, the posterior distribution of $y(x_{n+1})$ is given by (see Appendix for proof):

$$f(y(x_{n+1}) \mid \mathcal{H}_n) = \int f(y(x_{n+1}) \mid \Theta') \cdot f(\Theta' \mid \mathcal{H}_n) d\Theta'$$

$$= \int \int \mathcal{N}(x_{n+1}^\top \beta + \xi^{(z_{n+1})}(x_{n+1}), \tau^2) \underbrace{f(\xi^{(z_{n+1})}(x_{n+1}) \mid \Theta^{(2)}, \boldsymbol{\xi}_{1:n}, \boldsymbol{z}_{1:n}) f(\Theta, \boldsymbol{\xi}_{1:n}, \boldsymbol{z}_{1:n} \mid \mathcal{H}_n)}_{f(\Theta' \mid \mathcal{H}_n)}$$

$$d\xi^{(z_{n+1})}(x_{n+1}) d\{\Theta, \boldsymbol{\xi}_{1:n}, \boldsymbol{z}_{1:n}\}, \tag{8}$$

where $f(\cdot)$ denotes the probability density. According to this decomposition, given $\Theta'$, the distribution of $y(x_{n+1})$ is Gaussian: $y(x_{n+1})|\Theta' \sim \mathcal{N}(x_{n+1}^\top \beta + \xi^{(z_{n+1})}(x_{n+1}), \tau^2)$. Therefore, the (conditional) expected reward is $\mu^*(x_{n+1})|\Theta' = x_{n+1}^\top \beta + \xi^{(z_{n+1})}(x_{n+1})$.

**Thompson Sampling Implementation.** Therefore, the Thompson Sampling acquisition policy, as defined in Eq. (7), is implemented as follows:

(i) Draw a posterior sample $\hat{\Theta}' \sim f(\Theta' \mid \mathcal{H}_n)$ and (ii) Select the next location to evaluate:

$$x_{n+1} = \arg\max_{x \in \mathcal{X}} \hat{\mu}^*(x)|\hat{\Theta}' = \arg\max_{x \in \mathcal{X}} x^\top \hat{\beta} + \hat{\xi}^{(z_{n+1})}(x), \tag{9}$$

where the quantities with a hat (e.g., $\hat{\nu}, \hat{\beta}, \hat{K}_n, \{\hat{n}_j\}_{j=1}^{\hat{K}_n}$) denote posterior samples. The core challenge lies in drawing a posterior sample from $f(\Theta' \mid \mathcal{H}_n)$, particularly in sampling a path $\{\hat{\xi}^{(z_{n+1})}(x) : x \in \mathcal{X}\}$ from the posterior distribution. As shown in Eq. (8), we can achieve this by (i) first sampling a $\hat{\Theta}, \hat{\boldsymbol{\xi}}_{1:n}, \hat{\boldsymbol{z}}_{1:n}$ from the joint posterior $f(\Theta, \boldsymbol{\xi}_{1:n}, \boldsymbol{z}_{1:n} \mid \mathcal{H}_n)$ using a Gibbs sampler (see Appendix for details). (ii) Next, given $\hat{\Theta}, \hat{\boldsymbol{\xi}}_{1:n}, \hat{\boldsymbol{z}}_{1:n}$, we sample a path from the conditional posterior distribution $\{\xi^{(z_{n+1})}(x), x \in \mathcal{X}\} \mid \hat{\Theta}^{(2)}, \hat{\boldsymbol{\xi}}_{1:n}, \hat{\boldsymbol{z}}_{1:n}$. According to the Chinese restaurant process (CRP), a constructive representation of the Dirichlet process, a new upcoming data point either lies on one of the previously realized surfaces $\{\xi^{(j)}(\cdot)\}_{j=1}^{\hat{K}_n}$, or initiates a new surface drawn from the baseline distribution $G_0$ (as shown in the red dots in Figure 1). Based on this, we derive the following conditional distribution (see Appendix for proof):

$$f\big(\xi^{(z_{n+1})}(\cdot) \mid \hat{\Theta}^{(2)}, \hat{\boldsymbol{\xi}}_{1:n}, \hat{\boldsymbol{z}}_{1:n}\big) \sim \underbrace{P_n^{(0)} G_0(\cdot)}_{\text{Exploration}} + \underbrace{\sum_{j=1}^{\hat{K}_n} P_n^{(j)} \overbrace{\mathcal{GP}(\hat{\mu}_n^{(j)}(\cdot), \hat{\sigma}_n^2(\cdot))}^{\text{Kriging on } \xi^{(j)}}}_{\text{Exploitation}}, \qquad (10)$$

where $P_n^{(0)} = \frac{\hat{\nu}}{\hat{\nu}+n}$ and $P_n^{(j)} = \frac{\hat{n}_j}{\hat{\nu}+n}$. Each term $\mathcal{GP}(\hat{\mu}_n^{(j)}(\cdot), \hat{\sigma}_n^2(\cdot))$ corresponds to performing Kriging on an existing surface $\xi^{(j)}(\cdot)$, using past values $\xi^{(j)}(x_{1:n}) = (\xi^{(j)}(x_1), \cdots, \xi^{(j)}(x_n))$ realized on that surface. This follows the same calculation as in Eq. (1), with $\xi(x_{1:n})$ replaced by $\xi^{(j)}(x_{1:n})$. Efficient sampling from GP posterior $\mathcal{GP}(\hat{\mu}_n^{(j)}(\cdot), \hat{\sigma}_n^2(\cdot))$, as required here, has been widely explored in recent literature; see Wilson et al. [2020], Lin et al. [2023], Zhou [2025]. The pseudo code of our proposed $\infty$-GP-TS is shown in Algorithm 1.

**Exploration and Exploitation in Distributional Space.** As illustrated in Figure 1, Eq. (10) can be viewed as a structured trade-off between exploration and exploitation in the distributional space:

- **Exploitation:** With probability $P_n^{(j)} = \frac{\hat{n}_j}{\hat{\nu}+n}$, the new input $x_{n+1}$ is assigned to an existing surface $\xi^{(j)}(\cdot)$. In this case, the model performs Kriging inference using observations previously associated with that surface. This corresponds to exploiting empirical knowledge, resulting in conservative updates confined to the empirical distribution. Notably, the assignment probability is proportional to the number of observations $\hat{n}_j$ already associated with surface $j$. The more observations are assigned to a surface, the more likely it will be reused.

- **Exploration:** With probability $P_n^{(0)} = \frac{\hat{\nu}}{\hat{\nu}+n}$, a new surface is instantiated from the base GP prior $G_0$. Since no observations have been made on this surface, it represents functional-level exploration—expanding the surrogate model by admitting novel structures beyond those already observed.

As the number of observations $n$ increases, the exploration weight $P_n^{(0)}$ naturally diminishes, gradually **shifting focus from exploration toward exploitation** of well-supported surfaces. This hierarchical Bayesian mechanism thus equips Thompson Sampling with a principled way to adaptively balance exploration of new functional hypotheses and exploitation of learned structures, effectively broadening the surrogate model's support within the distribution space.

---

**Algorithm 1** Thompson Sampling with $\infty$-GP Surrogate Model ($\infty$-GP-TS)

---

**Input:** $\mathcal{H}_0$, the total number of iterations $N$.
**for** $n = 1$ to $N$ **do**
    **Step 1:** Sample $\{\hat{\Theta}, \hat{\boldsymbol{\xi}}_{1:n}\}$ from the posterior $f(\Theta, \boldsymbol{\xi}_{1:n} \mid \mathcal{H}_n)$ using Gibbs sampling (see the Appendix for details).
    **Step 2:** Sample a surface $\xi^{(z_{n+1})}$ from (10).
    **Step 3:** Evaluate the reward at the next candidate location $x_{n+1} = \arg\max_{x \in \mathcal{X}} x^\top \hat{\beta} + \hat{\xi}^{(z_{n+1})}(x)$. Observe the response $y(x_{n+1})$ and update $\mathcal{H}_{n+1} = \mathcal{H}_n \cup (x_{n+1}, y_{n+1})$.
    **Step 4:** Continue to the next iteration $n + 1$.
**end for**

---

# 5 Theoretical Analysis: Efficient Exploration in the Distributional Space

In this section, we show that by exploring the space of reward distributions, the proposed $\infty$-GP model can approximate a wide range of true reward models and achieve a near-minimax-optimal rate. Since the noisy reward is a stochastic process over $\mathcal{X}$, it is standard to characterize it in terms of its finite-dimensional distributions. Specifically, for any finite set of input locations $\boldsymbol{x}_{1:D} = \{x_1, \cdots, x_D\} \subset \mathcal{X}$ with $D \in \mathbb{N}^+$, we denote the corresponding $D$-dimensional joint density of the true reward values by $f^*(\cdot \mid \boldsymbol{x}_{1:D})$. Let $\boldsymbol{k} = (k_1, \ldots, k_D) \in \mathbb{N}_0^D$ be a multi-index. Define the corresponding mixed partial derivative operator of a function $f(y_1, \cdots, y_D)$ as $f^{(\boldsymbol{k})} := \partial^{k_1 + \cdots + k_D} f / \partial y_1^{k_1} \cdots \partial y_D^{k_D}$. For any $\alpha > 0$, $\lambda_0 \geq 0$, and any non-negative function $L : \mathbb{R}^D \to \mathbb{R}_{\geq 0}$, we define the locally Hölder class $\mathcal{C}^{\alpha, L, \lambda_0}(\mathbb{R}^D)$ as the set of all functions $f : \mathbb{R}^D \to \mathbb{R}$ that satisfy:(i) $f^{(\boldsymbol{k})}$ is finite for all multi-indices $\boldsymbol{k}$ such that $|\boldsymbol{k}| := \sum_{i=1}^D k_i \leq \lfloor \alpha \rfloor$; (ii) for every $x, y \in \mathbb{R}^D$, and $\boldsymbol{k}$ such that $|\boldsymbol{k}| = \lfloor \alpha \rfloor$, $\left| f^{(\boldsymbol{k})}(x + y) - f^{(\boldsymbol{k})}(x) \right| \leq L(x) e^{\lambda_0 \|y\|^2} \|y\|^{\alpha - \lfloor \alpha \rfloor}$.

**Assumption 1** *The true density $f^*(\cdot | \boldsymbol{x}_{1:D}) \in \mathcal{C}^{\alpha, L, \lambda_0}(\mathbb{R}^D)$ for some $\alpha > 0$, $\lambda_0 \geq 0$ and a non-negative function $L$ on $\mathbb{R}^D$ and satisfy $\mathbb{E}_{Y \sim f^*(\cdot | \boldsymbol{x}_{1:D})} \left( \frac{|f^{*(\boldsymbol{k})}(Y|x)|}{f^*(Y|x)} \right)^{\frac{2\alpha + \eta}{\sum_i^D k_i}} < \infty$ for any $\boldsymbol{k} \in \mathbb{N}_0^D$ satisfying $\sum_{i=1}^D k_i \leq \lfloor \alpha \rfloor$ and $\mathbb{E}_{Y \sim f^*(\cdot | \boldsymbol{x}_{1:D})} \left( \frac{L(Y)}{f^*(Y|\boldsymbol{x}_{1:D})} \right)^{\frac{2\alpha + \eta}{\alpha}} < \infty$ for some $\eta > 0$. Additionally, there are positive constant $a_1$, $a_2$, $a_3$, $\gamma$ such that*

$$f^*(y|\boldsymbol{x}_{1:D}) \leq a_1 \exp(-a_2 \|y\|^\gamma), \quad \|y\| > a_3. \tag{11}$$

Assumption 1 imposes mild regularity conditions on the true reward distribution. Apart from the tail condition in (11), the remaining smoothness assumptions are relatively weak. This assumption is satisfied by a broad class of distributions, including Weibull (with shape $k > 1$), Gaussian, Laplace, Gamma, and exponential distributions, as well as their finite mixtures of the form $\sum_{l=1}^L w_l(x) \psi_l(\theta_l)$, where each $\psi_l$ belongs to one of the aforementioned distribution families.

**Theorem 1 (Posterior Convergence of $\infty$-GP Model)** *Let $f^*(y \mid \boldsymbol{x}_{1:D})$ be the true reward distribution satisfying Assumption 1, $\Pi$ be the $\infty$-GP prior as defined in Eq. (4)-(6), and let $\Pi_n(\cdot \mid \boldsymbol{x}_{1:D})$ denote the posterior distribution based on $n$ i.i.d. evaluations at locations $\boldsymbol{x}_{1:D}$. Then there exists a sequence $\epsilon_n = n^{-\alpha/(2\alpha + D)}(\log n)^t$ for some constant $t > 0$, such that for any $M > 0$,*

$$\lim_{n \to \infty} \Pi_n \left( \{ f : \|f - f^*\|_1 > M\epsilon_n \} \mid \boldsymbol{x}_{1:D} \right) \to 0 \quad \text{almost surely under } f^*.$$

This result demonstrates that by efficiently exploring the distribution space, the posterior distribution of the $\infty$-GP model can contract around a broad class of true reward distributions at a near-minimax-optimal rate in $L_1$ distance. In contrast to classical GP regression, which requires strictly stronger functional class assumptions [Kanagawa et al., 2018, Theorem 5.1] than our method and assumes Gaussian noise, our framework allows much broader modeling flexibility.

# 6 Empirical Evaluation

We evaluate our method on ten benchmark tasks, including six synthetic functions, three real-world problems, and one LLM prompt optimization task. We particularly focus on BO tasks characterized by **non-stationary**, complex reward landscapes and **heavy-tailed** observation noise. Extensive results and ablations are provided in the Appendix.

**Synthetic Benchmarks.** We consider three popular synthetic test functions: Ackley, Rosenbrock, and StybTang [Xu et al., 2024]. To introduce more challenging scenarios, we consider non-stationary and heavy-tailed variants of these functions. In the heavy-tailed (HT) cases, all the functions are corrupted by Weibull-distributed noises. In the non-stationary (NS) setting, the base test functions are modulated by a trigonometric-exponential term of the form $f_{\text{NS}}(x) = (1 + \alpha \sin(x) e^x) \cdot f(x)$, which introduces non-stationarity across the domain.

**Real-world Benchmarks.** We consider three real-world benchmarks.

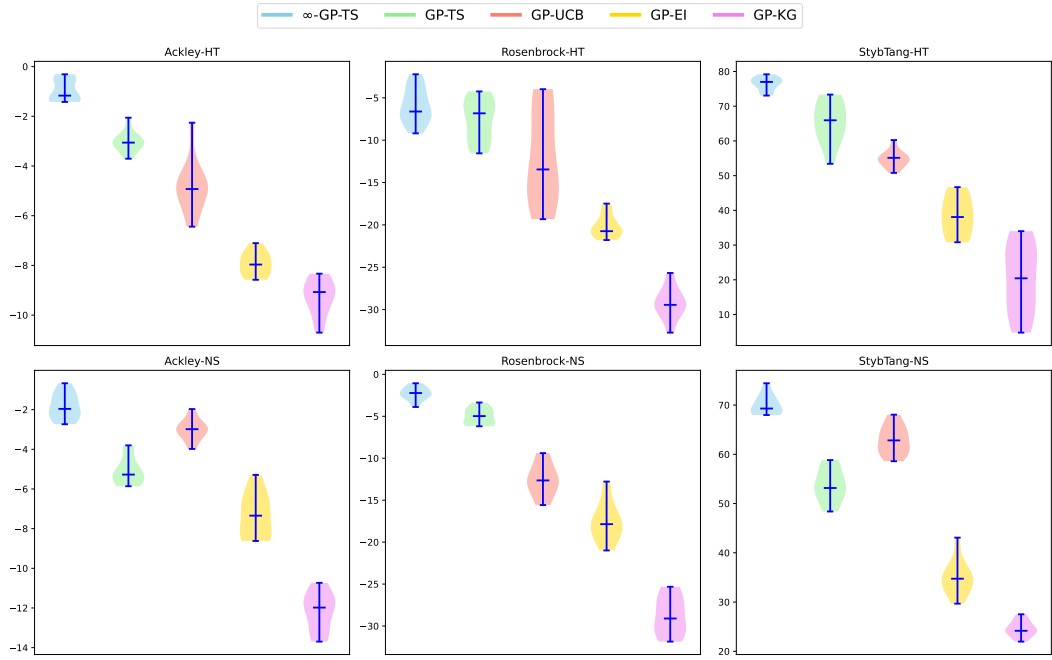

Figure 2: Results of synthetic benchmarks. The suffix HT stands for the function with heavy-tailed noises and NS stands for the non-stationary scenarios.

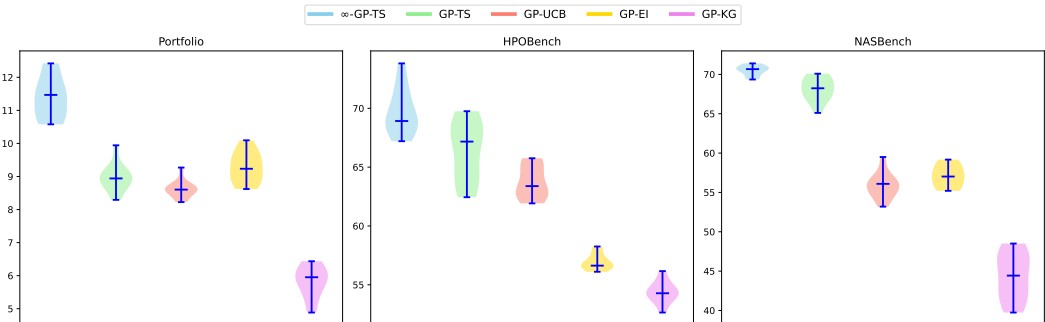

Figure 3: Results of real-world benchmarks.

- **Portfolio** [Cakmak et al., 2020] is a benchmark of tuning the hyperparameters of a trading strategy to maximize returns. The stock data with heavy-tailed nature is generated from a realistic financial simulator, CVXPortfolio.

- **HPOBench** [Eggensperger et al., 2021] provides standard hyperparameter tuning tasks for ML models, and we use the MLP task in the HPOBench in our experiment.

- **NASBench201** [Dong and Yang, 2020] is a neural network search task on CIFAR100. The search space of hyperparameters and model architecture is considered non-stationary. Therefore, we use these tasks for our evaluation of real-world non-stationary cases.

**Prompt optimization for language understanding.** We conduct prefix prompt optimization for seven language understanding datasets, including sentiment classification (SST-2 [Socher et al., 2013]), SST-5 [Socher et al., 2013], MR [Pang and Lee, 2005], CR [Hu and Liu, 2004]), topic classification (AG's News[Zhang et al., 2015], TREC[Voorhees and Tice, 2000]) and subjectivity classification (Subj [Pang and Lee, 2004]). We use the Alpaca-7B model [Taori et al., 2023] as the frozen language model, and optimize continuous prefix embeddings to guide the model toward better task performance. Due to the high dimensions of the decision variable, we use Uniform Manifold Approximation and Projection (UMAP) [McInnes et al., 2018] to reduce the dimension.

| Method | SST-2 | CR | MR | SST-5 | AG's News | TREC | Subj | Avg. |
|---|---|---|---|---|---|---|---|---|
| GP-TS | 93.87 | 91.44 | 89.85 | 49.91 | 72.81 | 45.29 | 65.18 | 72.61 |
| GP-UCB | 92.68 | 90.20 | 89.98 | 49.86 | 71.76 | 43.27 | 62.70 | 71.49 |
| GP-EI | 91.82 | 89.75 | 87.69 | 48.26 | 60.83 | 42.09 | 59.70 | 68.59 |
| GP-KG | 90.05 | 88.38 | 85.90 | 42.89 | 45.98 | 36.20 | 49.75 | 62.74 |
| $\infty$-GP-TS | **96.57** | **92.52** | **91.51** | **52.58** | **75.34** | **46.23** | **68.55** | **74.76** |

Table 1: Performance comparison on seven language understanding datasets.

**Baselines.** We compare our methods with the following baselines based on GP surrogate model. **GP-TS** [Chowdhury and Gopalan, 2017]: Following the TS policy defined in Eq. (7), except that the posterior is constructed under a standard GP model for $\mu^*(x)$. **GP-UCB** [Srinivas et al., 2009]: Selects the location $x_{n+1} = \arg\max_{x \in \mathcal{X}} \mu_n(x) + \beta_n^{1/2}\sigma_n(x)$, where $\beta_n$ is a time-dependent exploration parameter. **GP-EI** [Ament et al., 2023]: Selects $x_{n+1} = \arg\max_{x \in \mathcal{X}} \mathbb{E}\left[\max\{0, \mu^*(x) - \mu^{\text{best}}\}\right]$, where the expectation is taken under the posterior distribution of $\mu^*(x)$. **GP-KG** [Wu and Frazier, 2016]: Selects the point whose evaluation is expected to most improve the posterior estimate of the maximum value of $\mu^*(x)$. **Non-stationary GP-UCB/TS/EI/KG**: In non-stationary tasks and the Prompt optimization task, all baseline methods adopt input warping techniques in Snoek et al. [2014] to mitigate GP misspecification. **Truncated-GP-UCB**: In heavy-tailed noise tasks, GP-UCB adopts the truncation technique in Chowdhury and Gopalan [2019].

**Performance.** We report the final optimization outcome of the synthetic and real-world experiment in Figure 2 and 3, respectively (runtime regret results are provided in the Appendix). The result of each task is reported from 10 repetitive runs with different seeds. In the synthetic experiment, $\infty$-GP consistently outperforms the baselines in heavy-tailed and non-stationary settings. The GP-TS and GP-UCB have similar performance against each other. The GP-EI performs poorly in both scenarios. The GP-KG has the worst results, and the final performance fluctuates heavily.

In the portfolio task, we report the returns of the strategy. Stock prices often exhibit sharp fluctuations, making portfolio optimization a suitable testbed for evaluating performance under heavy-tailed environments. Our method achieves the highest return among others, while the GP-EI has the second-best return. In the HPOBench and the NASBench, the performance gap between our method and baselines is close, but our method has a smaller fluctuation with different seeds, showing better robustness than others.

The language model in the prompt optimization task is Alphaca-7b. Table 1 reports the average final accuracy across three independent runs. The proposed $\infty$-GP outperforms all GP-based baselines on average, achieving the highest mean accuracy (74.76), and ranking first on every dataset. Compared to GP-TS and GP-UCB,

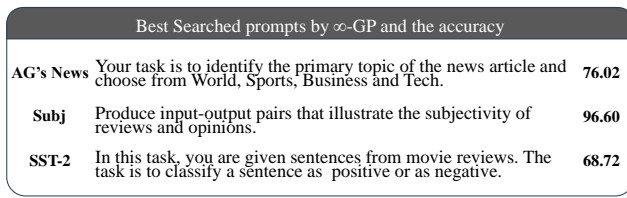

Figure 4: Qualitative examples of best searched prompts

$\infty$-GP offers notable gains on complex tasks such as SST-5 and Subj, where the reward landscape is highly non-stationary. GP-EI and GP-KG perform significantly worse, consistent with their known sensitivity to noisy or irregular reward signals. These results demonstrate the robustness of $\infty$-GP in prompt optimization tasks under distributional uncertainty.

# 7 Discussion

This work introduces the $\infty$-GP model and its integration with TS, providing a principled mechanism to automatically balance exploration and exploitation in the space of reward distributions. Empirically, the proposed method demonstrates strong performance in a wide range of challenging settings where classical GP-based BO methods often fail. Nonetheless, we do not systematically evaluate the performance of $\infty$-GP in high-dimensional BO tasks. That said, as shown in Eq. (10) and Algorithm 1, our approach can be interpreted as a finite mixture of traditional GP-TS. This structure is orthogonal to many recent advances in high-dimensional GP-based BO, and hence, those techniques can be readily incorporated into our framework. We leave the exploration of such extensions to future work.

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

## A Proofs

### A.1 Proof of Eq. (8) and Eq. (10)

We adopt the bracket notation system used in Gelfand and Smith [1990], where $[Y \mid X]$ denotes the conditional density of $Y$ given $X$, and $[Y]$ denotes the marginal density of $Y$.

The predictive reward distribution for any unexplored solution $x_{n+1}$ is given by

$$[y(x_{n+1})|\mathcal{H}_n] = \int \underbrace{[y(x_{n+1})|\Theta^{(1)}, \xi^{(z_{n+1})}(x_{n+1})]}_{A: \text{ Gaussian}} \underbrace{[\Theta^{(1)}, \xi^{(z_{n+1})}(x_{n+1})|\mathcal{H}_n]}_{B+C}$$

$$= \int \int \underbrace{[y(x_{n+1})|\Theta^{(1)}, \xi^{(z_{n+1})}(x_{n+1})]}_{A: \text{ Gaussian}} \underbrace{[\xi^{(z_{n+1})}(x_{n+1})|\boldsymbol{\xi}_{1:n}(x_{1:n}), \boldsymbol{z}_{1:n}, \Theta^{(2)}]}_{B} \qquad (12)$$

$$\underbrace{[\Theta, \boldsymbol{\xi}_{1:n}(x_{1:n}), \boldsymbol{z}_{1:n}|\mathcal{H}_n]}_{C}$$

Notice that, when performing computations involving stochastic processes over $\mathcal{X}$, in practice, we are working with their finite-dimensional distributions, i.e., the joint distribution over a finite subset of $\mathcal{X}$. Therefore, the term $\boldsymbol{\xi}_{1:n}$ becomes $\boldsymbol{\xi}_{1:n}(x_{1:n}) = \{\xi^{(j)}(x_i)\}_{i=1,\cdots,n \; j=1,\cdots,K_n}$, which represents the $K_n \times n$ table of values on the realized surfaces corresponding to the observed locations $x_{1:n}$.

In the first line of Eq. (12), conditioned on the new surface $\xi^{(z_{n+1})}(x_{n+1})$ and first-layer hyper parameter $\Theta^{(1)} = (\beta, \tau^2)$, the distribution $[y(x_{n+1})|\Theta^{(1)}, \xi^{(z_{n+1})}(x_{n+1})]$ is $\mathcal{N}(x_{n+1}^\top\beta + \xi^{(z_{n+1})}(x_{n+1}), \tau^2)$, according to Eq. (4). The remaining Term "B+C" is $[\Theta'|\mathcal{H}_n]$. In the second line of Eq. (12), the posterior distribution $[\Theta^{(1)}, \xi^{(z_{n+1})}|\mathcal{H}_n]$, i.e. the Term "B+C", is decomposed into Term B and Term C. Term C represents the posterior distribution of the hyperparameters and realized surfaces, which requires inference through Gibbs sampling. The proof of Eq. (8) is now complete.

We next proceed to establish Eq. (10). The Term B represents the prediction of the new surface $\xi^{(z_{n+1})}$ based on the previously realized surfaces $\boldsymbol{\xi}_{1:n} = (\xi^{(1)}, \ldots, \xi^{(K_n)})$ and their associated assignment labels $\boldsymbol{z}_{1:n}$. Conditioned on $\boldsymbol{\xi}_{1:n} = (\xi^{(1)}, \ldots, \xi^{(K_n)})$ and $\boldsymbol{z}_{1:n}$, and marginalizing out the random measure $\{G_x : x \in \mathcal{X}\}$, the distribution of the latent surface index $z_{n+1}$ follows the urn scheme of the Dirichlet process [Blackwell and MacQueen, 1973]. Specifically, the new point either reuses one of the existing surfaces (solid lines in Figure 1) or initiates a new surface drawn from the base measure $G_0 = \mathcal{GP}(0, \sigma^2\rho_\phi(\cdot, \cdot))$ (dashed lines in Figure 1). Formally:

$$\underbrace{[\xi^{(z_{n+1})} \mid \boldsymbol{\xi}_{1:n}, \boldsymbol{z}_{1:n}, \Theta^{(2)}]}_{\text{Urn scheme}}$$

$$= \int [\xi^{(z_{n+1})} \mid \boldsymbol{\xi}_{1:n}, \boldsymbol{z}_{1:n}, \Theta^{(2)}, G] [G] \qquad (13)$$

$$\sim \frac{\nu}{\nu + n}G_0 + \sum_{j=1}^{K_n} \frac{n_j}{\nu + n}\delta_{\xi^{(j)}},$$

where $n_j$ denotes the number of data points currently assigned to surface $\xi^{(j)}$. The more points are assigned to a surface, the more likely it is to be reused. Then, after determining which surface $\xi^{(z_{n+1})}$ the new location $x_{n+1}$ will be realized on, we perform Kriging on that surface to predict $\xi^{(z_{n+1})}(x_{n+1})$ using previously realized values $\boldsymbol{\xi}_{1:n}(x_{1:n})$. Therefore, the Term B can be decomposed as

$$[\xi^{(z_{n+1})}(x_{n+1})|\boldsymbol{\xi}_{1:n}(x_{1:n}), \boldsymbol{z}_{1:n}, \Theta^{(2)}]$$

$$= \int \underbrace{[\xi^{(z_{n+1})}(x_{n+1})|\boldsymbol{\xi}_{1:n}(x_{n+1}), \boldsymbol{z}_{1:n}, \Theta^{(2)}]}_{\text{Step (a): Urn scheme (13)}} \underbrace{[\boldsymbol{\xi}_{1:n}(x_{n+1})|\boldsymbol{\xi}_{1:n}(x_{1:n}), \Theta^{(2)}]}_{\text{Step (b): kriging}}, \qquad (14)$$

where the urn scheme term follows (13) by evaluating $\xi^{(z_{n+1})}$ and $\boldsymbol{\xi}_{1:n}$ at location $x_{n+1}$. The Kriging term is given by (recall that $\boldsymbol{\xi}_{1:n} = \{\xi^{(j)}\}_{j=1}^{K_n}$)

$$\underbrace{[\xi^{(j)}(x_{n+1})|\boldsymbol{\xi}_{1:n}(x_{1:n}), \Theta^{(2)}]}_{\text{kriging}} \sim \mathcal{GP}(\mu_n^{(j)}(x_{n+1}), \sigma_n^2(x_{n+1})), \qquad (15)$$

where $\mu_n^{(j)}(x_{n+1}) = \Sigma_0(x_{n+1}, x_{1:n})\Sigma_0(x_{1:n}, x_{1:n})^{-1}\xi^{(j)}(x_{1:n})$, and $\sigma_n^2(x_{n+1})$ is given by Eq. (3). Therefore, combing Eq. (14) and Eq. 15, we have

$$f\big(\xi^{(z_{n+1})}(\cdot) \mid \hat{\Theta}^{(2)}, \hat{\boldsymbol{\xi}}_{1:n}, \hat{\boldsymbol{z}}_{1:n}\big) \sim \underbrace{P_n^{(0)} G_0(\cdot)}_{\text{Exploration}} + \underbrace{\sum_{j=1}^{\hat{K}_n} P_n^{(j)} \overbrace{\mathcal{GP}(\hat{\mu}_n^{(j)}(\cdot), \hat{\sigma}_n^2(\cdot))}^{\text{Kriging on } \xi^{(j)}}}_{\text{Exploitation}}, \tag{16}$$

which concludes the proof.

## A.2   Proof of Theorem 1

For any finite set of input locations $x_{1:D} = \{x_1, \cdots, x_D\} \subset \mathcal{X}$ with $D \in \mathbb{N}^+$, we consider the corresponding $D$-dimensional joint density of the true reward values by $f^*(\cdot \mid x_{1:D})$.

Notice that, focusing on fixed $x_{1:D}$, the $\infty$-GP model can be viewed as a DP location mixture of normals prior $\Pi$, which is the distribution of a random distribution $f_{DP_G,\Sigma} = \int \phi_\Sigma(x - z) DP_G(dz)$. $\phi_\Sigma$ is the normal density with mean zero and covariance $\Sigma$. $DP_G$ is a Dirichlet process with baseline distribution $G$, which is a $D$-dimension Gaussian distribution, with squared exponential kernel $\rho_\phi(x, x') = \exp\left(-\sum_{k=1}^d \phi_k(x^{(k)} - x'^{(k)})^2\right)$ (i.e. the kernel of baseline of $\infty$-GP) and zero mean. In our model, we assume the covariance matrix $\Sigma \in \mathbb{R}^{D \times D}$ to be diagonal with independent inverse-gamma priors on each variance component:

$$\Sigma = \begin{bmatrix} \tau_1^2 & 0 & \cdots & 0 \\ 0 & \tau_2^2 & \cdots & 0 \\ \vdots & \vdots & \ddots & \vdots \\ 0 & 0 & \cdots & \tau_D^2 \end{bmatrix},$$

with $\tau_i^2 \sim \text{Inv-Gamma}(a_\tau, b_\tau)$ independently for $i = 1, \ldots, D$.

Let $|G| = G(\mathbb{R}^D)$ and $\bar{G} = G/|G|$. Let $\text{eig}_j$ be the $j$-th eigenvalue. We first present the following lemma.

### A.2.1   Lemma 1

**Lemma 1** *There exist positive constants $a_1$-$a_8$.*

$$1 - \bar{G}([-x, x]^D) \le a_1 \exp(-a_2 x^{a_3}), \tag{17}$$

*and*

$$\mathbb{P}(\Sigma : eig_D(\Sigma^{-1}) \ge x) \le a_4 \exp(-a_5 x^{a_6}). \tag{18}$$

*for sufficiently large $x > 0$, and*

$$\mathbb{P}(\Sigma : eig_1(\Sigma^{-1}) < x) \le a_7 X^{a_8} \tag{19}$$

*for sufficiently small $x > 0$. Moreover, there exist $a_9$ such that for any $0 < s_1 \le \cdots \le s_D$ and $t \in (0, 1)$,*

$$\mathbb{P}(\Sigma : s_j < eig_j(\Sigma^{-1}) < s_j(1 + t), j = 1, \cdots, D) \ge a_9 s_1^{a_{10}} t^{a_{11}} \exp(-a_{12} s_d). \tag{20}$$

*Proof*: Equation (17) follows from the exponential tail behavior of the Gaussian base measure $G$. To prove Equation (18), recall that $\Sigma = \text{diag}(\tau_1^2, \ldots, \tau_D^2)$ and hence $\text{eig}_D(\Sigma^{-1}) = \max_i \left\{\frac{1}{\tau_i^2}\right\}$. Since $\tau_i^2 \sim \text{Inv-Gamma}(a_\tau, b_\tau)$, we use the left-tail bound for inverse-gamma random variables:

$$\mathbb{P}\left(\tau_i^2 \le \frac{1}{x}\right) \le C_1 \exp(-C_2 x), \quad \text{as } x \to \infty.$$

Applying a union bound over $i = 1, \ldots, D$, we obtain:

$$\mathbb{P}\left(\text{eig}_D(\Sigma^{-1}) \ge x\right) \le D \cdot C_1 \exp(-C_2 x) = a_4 \exp(-a_5 x^{a_6}).$$

For Equation (19), we use the right-tail of the inverse-gamma distribution:

$$\mathbb{P}\left(\tau_i^2 \geq \tfrac{1}{x}\right) = \mathbb{P}\left(\frac{1}{\tau_i^2} \leq x\right) \leq C_3 x^{a_\tau}, \quad \text{as } x \to 0^+.$$

Therefore,

$$\mathbb{P}\left(\mathrm{eig}_1(\Sigma^{-1}) < x\right) = \mathbb{P}\left(\frac{1}{\tau_i^2} < x \text{ for some } i\right) \leq D \cdot C_3 x^{a_\tau} = a_7 x^{a_8}.$$

Finally, for the lower bound in Equation (20), note that due to the independence of $\tau_i^2$, the eigenvalues of $\Sigma^{-1}$ are also independent. Thus, for each $j$, we consider the event:

$$\mathrm{eig}_j(\Sigma^{-1}) \in [s_j, s_j(1+t)],$$

which translates to $\tau_j^2 \in \left[\frac{1}{s_j(1+t)}, \frac{1}{s_j}\right]$.

The inverse-gamma distribution has a continuous, strictly positive density on compact intervals bounded away from zero. Therefore, for each $j$,

$$\mathbb{P}\left(\frac{1}{\tau_j^2} \in [s_j, s_j(1+t)]\right) \geq c_j s_j^{-b} t,$$

for some constants $c_j, b > 0$. Multiplying across $j = 1, \ldots, D$, we get:

$$\mathbb{P}\left(\mathrm{eig}_j(\Sigma^{-1}) \in [s_j, s_j(1+t)] \ \forall j\right) \geq a_9 s_1^{a_{10}} t^{a_{11}} \exp(-a_{12} s_D),$$

where we used the fact that inverse-gamma densities decay exponentially on the left (i.e., for small $\tau_j^2$), which leads to the $\exp(-a_{12} s_D)$ term.

### A.2.2 Lemma 2

**Lemma 2** *[Ghosal and van der Vaart [2007]]*

$$\lim_{n \to \infty} \Pi_n \left(\{f : \|f - f^*\|_1 > M\epsilon_n\} \mid \boldsymbol{x}_{1:D}\right) \to 0 \quad \text{almost surely under } f^*$$

*whenever there exist positive constants $c_1, c_2, c_3, c_4$, a sequence of positive numbers $(\tilde{\epsilon}_n)_{n \geq 1}$ with $\tilde{\epsilon}_n \leq \epsilon_n$ and $\lim_{n \to \infty} n\tilde{\epsilon}_n^2 = \infty$, and a sequence of compact subsets $(\mathcal{F}_n)_{n \geq 1}$ of probability densities satisfying*

$$\log N(\epsilon_n, \mathcal{F}_n, \rho) \leq c_1 n \epsilon_n^2, \tag{21}$$

$$\Pi(\mathcal{F}_n^c) \leq c_3 e^{-(c_2+4)n\tilde{\epsilon}_n^2}, \tag{22}$$

$$\Pi\{f : \mathcal{K}(f^*, \tilde{\epsilon}_n) \leq \tilde{\epsilon}_n^2, \} \geq c_4 e^{-c_2 n \tilde{\epsilon}_n^2}. \tag{23}$$

*then the posterior contracts at the rate $\epsilon_n$ around $f_0$ in the metric $\rho$.*

Here, $\mathcal{K}(f^*, \epsilon)$ is the Kullback–Leibler ball around $f^*$ of size $\tilde{\epsilon}_n$, defined as

$$\mathcal{K}(f^*, \epsilon) = \left\{f : \int f^* \log\left(\frac{f^*}{f}\right) < \epsilon^2, \ \int f^* \log^2\left(\frac{f^*}{f}\right) < \epsilon^2\right\}.$$

$N(\epsilon_n, \mathcal{F}_n, \rho)$ is the $\epsilon_n$-covering number of $\mathcal{F}_n$ under metric $\rho$.

### A.2.3 Proof of Theorem 1

Now we prove Theorem 1 by verifying conditions (21), (22) and (23), respectively. We construct a sieve

$$\mathcal{Q}(a, M, J, \epsilon) \triangleq \left\{ f_{DP_G, \Sigma} : \begin{array}{l} DP_G = \sum_{j=1}^\infty w_j \delta_{\xi_j(x_{1:D})}, \ \xi_j(x_{1:D}) \in [-a, a]^D \text{ for } j < J; \\ \sum_{j>J} w_j < \epsilon, \sigma_0^2 \leq \mathrm{eig}_j(\Sigma) < \sigma_0^2 \left(1 + \frac{\sigma^2}{D}\right)^M, \ \forall j = 1, \ldots, D \end{array} \right\}.$$

Then, according to [Shen et al., 2013, Theorem 5] and Lemma 1, conditions (21) and (22) hold for $\epsilon_n = n^{-\gamma'}(\log n)^t$, $\tilde{\epsilon}_n = n^{-\gamma'}(\log n)^{t_0}$, $J = \lfloor \frac{n\epsilon_n^2}{\log n} \rfloor$, $M = a^{a_2}$, when fixing $\gamma' \in (0, 1/2)$ and $t > t_0 \geq \frac{D+1}{2}$.

To verify condition (23), we follow the prior thickness argument in Shen et al. [2013, Theorem 4]. Under Assumption 1, the true density $f^*$ belongs to a locally Hölder smooth class with exponential tail decay. Theorem 4 guarantees that there exists a discrete mixture density $p_{F,\Sigma}$ such that both the Kullback–Leibler divergence and its second moment between $f^*$ and $p_{F,\Sigma}$ are bounded by $C\tilde{\epsilon}_n^2$ for some constant $C > 0$. Moreover, the Dirichlet process prior assigns exponentially non-negligible mass to such mixtures via a partitioning and stick-breaking construction. The prior on the covariance matrix $\Sigma$, as discussed in Lemma 1, also places sufficient mass on the eigenvalue shell corresponding to the mixture approximation. Combining these results, let $\tilde{\epsilon}_n^2 = n^{-\alpha/(2\alpha+D)}(\log n)^t$, for any $t \geq \frac{D(1+1/\gamma+1/\alpha)+1}{2+D+\alpha}$, according to Shen et al. [2013, Theorem 4], we obtain

$$\Pi\left\{ f : \mathrm{KL}(f^*, f) \leq \tilde{\epsilon}_n^2, \ \mathrm{Var}_{f^*} \log \frac{f^*}{f} \leq \tilde{\epsilon}_n^2 \right\} \geq c_4 e^{-c_2 n \tilde{\epsilon}_n^2},$$

for some constants $c_2, c_4 > 0$, thus verifying condition (23). Finally, according to Lemma (2), the proof is concluded.

## B   Gibbs Sampling Algorithm

### B.1   Prior Specification

We adopt a fully-Bayesian treatment for hyperparameters $\Theta = \{\beta, \nu, \tau, \sigma^2, \phi\} = \{\Theta^{(1)}, \Theta^{(2)}\}$, where the fist-layer parameters $\Theta^{(1)} = \{\beta, \tau\}$ and the second-layer parameters $\Theta^{(2)} = \{\nu, \sigma^2, \phi\}$. Specifically, the priors on hyperparameters $\Theta$ are given by

$$\beta, \tau^2 \sim N_p(\beta_0, \Sigma_\beta) \times IGamma(a_\tau, b_\tau), \tag{24}$$

$$\sigma^2 \sim IGamma(a_\sigma, b_\sigma), \tag{25}$$

$$\phi \sim U([0, b_\phi]^d), \tag{26}$$

$$\nu \sim Gamma(a_\nu, b_\nu), \tag{27}$$

where $\beta$ has a Gaussian prior, $\tau^2$ and $\sigma^2$ have inverse Gamma prior, $\phi$ has a uniform prior on $(0, b_\phi]$ and $\nu$ has a Gamma prior.

**Selection of Hyperparameters in the Prior**   we set $a_\tau = a_\sigma = 2$ and thus the mean of the prior is $b_\tau$ and $b_\sigma$. Therefore, prior information about the variance (for example, a rough estimation of the variance and the mean) can be incorporated.

$\phi$: $\phi$ has a uniform prior on $(0, b_\phi]$. If the distance between two locations $x_1$ and $x_2$, $\|x_1 - x_2\| > \frac{3}{\phi}$, their correlation will decrease to less than 0.05. Therefore, we set $\frac{3}{b_\phi} = d \max\|x_1 - x_2\|$ with a small $d$ (e.g. 0.01).

$\beta_0$ and $\Sigma_\beta$: we set $\beta_0 = [1, \cdots, 1] \in \mathbb{R}^d$ and set $\Sigma_\beta = I_{d \times d}$.

### B.2   Gibbs Sampling

To perform posterior inference under our fully-Bayesian model, we employ the Gibbs sampling algorithm. Gibbs sampling is a Markov Chain Monte Carlo (MCMC) method designed to sample from a complex joint posterior distribution by iteratively sampling from each of its full conditional distributions. This approach is particularly well-suited for our hierarchical $\infty$-GP model, where the full conditionals of most parameters have closed-form expressions or can be easily computed. To ensure computational tractability, we adopt a truncation approximation to the stick-breaking representation of the Dirichlet process. Specifically, we fix the number of mixture components to a finite value $L$.

**Algorithm 2** Gibbs Sampling for $\{\Theta, \boldsymbol{\xi_{1:n}}(x_{1:n})\}$

---

1: **Input:** Initialization value of $\{\Theta, \boldsymbol{\xi_{1:n}}(x_{1:n})\}$, data $\mathcal{H}_n$, MCMC iteration number $B$, truncation number $L$

2: **Output:** Samples of $[\Theta, \boldsymbol{\xi_{1:n}}(x_{1:n})|\mathcal{H}_n]$, including $\boldsymbol{\xi_{1:n}}(x_{1:n})$, $(w_1, \ldots, w_L)$, $(z_1, \ldots, z_n)$, and $\Theta = \{\beta, \nu, \tau, \sigma^2, \phi\}$.

3: **for** each iteration $b = 1, \ldots, B$ **do**

4:     **Step 1: Sampling $\boldsymbol{\xi_{1:n}}(x_{1:n})$:**

5:     **for** each $l = 1, \ldots, L$ **do**

6:         Compute $\mathcal{I}^l(\boldsymbol{z_{1:n}}) = \mathrm{diag}(\mathbb{I}_{\{z_1=l\}}, \ldots, \mathbb{I}_{\{z_n=l\}})$

7:         Sample $\xi_l(x_{1:n})$ from the distribution $\xi_l(x_{1:n}) \sim \mathcal{N}\left(\frac{1}{\tau^2}\Lambda^l \mathcal{I}^l \left(y(x_{1:n}) - x_{1:n}^\top\beta\right), \Lambda^l\right)$,
    where $\Lambda^l = \left(\Sigma_0^{-1} + \tau^{-2}\mathcal{I}^l\right)^{-1}$ and $\Sigma_0 = \sigma^2 H(\phi)$

8:     **end for**

9:     **Step 2: Sampling Weights $(w_1, \ldots, w_L)$:**

10:     **for** each $l = 1, \ldots, L$ **do**

11:         Draw $V_l \sim \mathrm{Beta}(1 + M_l, \nu + \sum_{j=l+1}^{L} M_j)$, where $M_j = \#\{i : z_i = j\}$

12:         Compute weights by $w_1 = V_1$, $w_l = (1 - V_1)(1 - V_2)\ldots(1 - V_{l-1})V_l$, $w_L = 1 - \sum_{l=1}^{L-1} w_l$

13:     **end for**

14:     **Step 3: Sampling Latent Labels $(z_1, \ldots, z_n)$:**

15:     **for** each observation $x_i$ **do**

16:         Sample $z_i$ from $\{1, \ldots, L\}$ with probabilities proportional to

$$P(z_i = j) \propto w_j \exp\left\{-\frac{1}{2\tau^2}\left(y(x_i) - x_i^\top\beta - \xi_j(x_i)\right)^2\right\}.$$

17:     **end for**

18:     **Step 4: Sampling Hyper-Parameters $\Theta = \{\beta, \nu, \tau, \sigma^2, \phi\}$:**

19:     **(4a) Sampling $\nu$:** $\nu \sim \mathrm{Gamma}(a_\nu + n - 1, b_\nu - \log(w_L))$

20:     **(4b) Sampling $\beta$:**

$$\beta \sim \mathcal{N}(\beta_0^n, \Sigma_\beta^n)$$

    where $\Sigma_\beta^n = \left(\Sigma_\beta^{-1} + \tau^{-2}x_{1:n}^\top x_{1:n}\right)^{-1}$, $\beta_0^n = \Sigma_\beta^n\left(\Sigma_\beta^{-1}\beta_0 + \tau^{-2}X_{1:n}^\top\bar{y}(x_{1:n})\right)$ and $\bar{y}(x_i) = y(x_i) - \xi^{(z_i)}(x_i)$.

21:     **(4c) Sampling $\tau^2$:**

$$\tau^2 \sim \mathrm{InvGamma}(a_\tau^n, b_\tau^n)$$

    where $a_\tau^n = a_\tau + \frac{n}{2}$, $b_\tau^n = b_\tau + \frac{1}{2}\sum_{i=1}^{n}\left(\bar{y}(x_i) - x_i^\top\beta\right)^2$.

22:     **(4d) Sampling $\sigma^2$:**

$$\sigma^2 \sim \mathrm{IGamma}(a_\sigma^n, b_\sigma^n)$$

    where $a_\sigma^n = a_\sigma + \frac{nL}{2}$, $b_\sigma^n = b_\sigma + \frac{1}{2}\sum_{l=1}^{L}\left(\xi_l(x_{1:n})\right)^\top \rho_\phi^{-1}(x_{1:n}, x_{1:n})\xi_l(x_{1:n})$.

23:     **(4e) Sampling $\phi$:** $\phi$ is sampled from a grid of values $\{\phi_1, \phi_2, \ldots, \phi_M\}$ with probabilities proportional to

$$P(\phi_m) \propto \frac{1}{[\det(\rho_{\phi_m}(x_{1:n}, x_{1:n}))]^{L/2}} \exp\left(-\frac{1}{2\sigma^2}\sum_{l=1}^{L}\left(\xi_l(x_{1:n})\right)^\top \rho_{\phi_m}^{-1}(x_{1:n}, x_{1:n})\xi_l(x_{1:n})\right).$$

24: **end for**

---

# C Experimental Details, More Experiment Results and Ablations

## C.1 More Experiment Results

In this subsection, we provide additional plots that visualize the evolution of instant regret over time for all benchmark tasks, where instant regret is defined as the gap between the reward obtained and the global optimum. These plots offer a clearer comparison of the performance dynamics across different algorithms.

### C.1.1 Synthetic Benchmarks

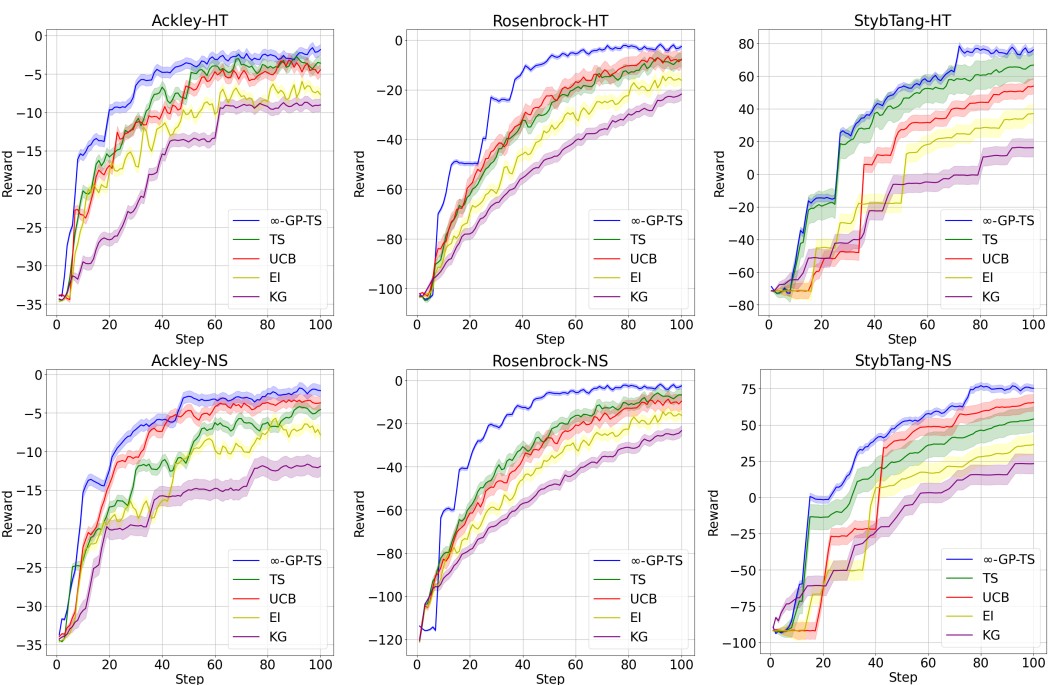

Figure 5: Cumulative regret in synthetic benchmarks

Figure 5 shows the evolution of instant regret over time across various synthetic benchmark functions. The compared methods—TS, UCB, EI, and KG—are all implemented using standard Gaussian Process surrogate models. Overall, we observe that $\infty$-GP-TS achieves the best performance in settings with nonstationary or heavy-tailed reward structures, significantly outperforming classical GP-based approaches. In the early iterations (e.g., the first 10 steps), the performance of $\infty$-GP-TS is initially lower due to its stronger exploration behavior. However, it quickly adapts and achieves lower regret as the optimization progresses, demonstrating its ability to efficiently learn complex reward landscapes.

### C.1.2 Real-world Benchmark

Figure 6 shows the runtime performance (instant regret) on three real-world benchmarks: Portfolio, HPOBench, and NASBench. Across all tasks, NBO consistently outperforms other methods, achieving faster convergence and lower regret. In the Portfolio task, NBO quickly surpasses all baselines; in HPOBench and NASBench, it maintains a clear lead throughout the optimization process. These results highlight NBO's strong generalization ability and its effectiveness in complex, real-world settings.

## C.2 More Experiment Details

In this subsection, we present more experiment details, including the synthetic benchmarks, real-world benchmarks and the prompt optimization problem.

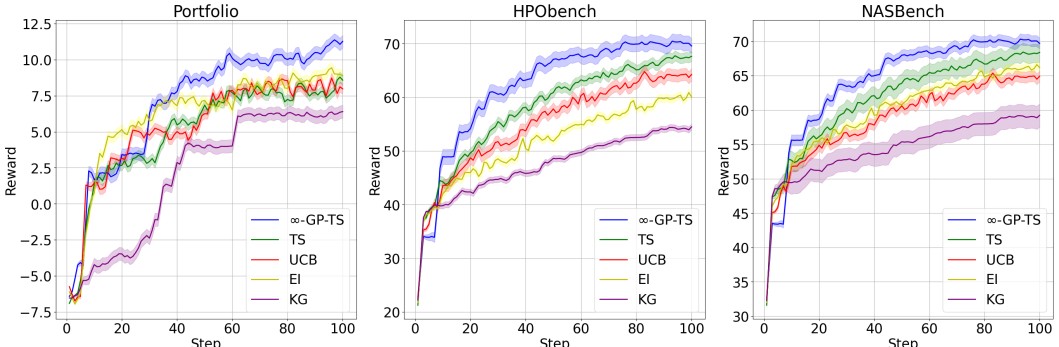

Figure 6: Cumulative regret in real-world benchmarks

### C.2.1 Benchmark Function Definitions

We consider three popular synthetic test functions commonly used in global optimization benchmarks: the Ackley function, the Rosenbrock function, and the Styblinski–Tang (StybTang) function. These functions are defined over a $d$-dimensional input vector $x = (x_1, \ldots, x_d) \in \mathbb{R}^d$.

**Ackley Function.**

$$f_{\text{Ackley}}(x) = -a \exp\left(-b\sqrt{\frac{1}{d}\sum_{i=1}^{d} x_i^2}\right) - \exp\left(\frac{1}{d}\sum_{i=1}^{d} \cos(cx_i)\right) + a + \exp(1),$$

with default parameters $a = 20$, $b = 0.2$, and $c = 2\pi$.

**Rosenbrock Function.**

$$f_{\text{Rosenbrock}}(x) = \sum_{i=1}^{d-1} \left[100(x_{i+1} - x_i^2)^2 + (x_i - 1)^2\right].$$

**Styblinski–Tang Function.**

$$f_{\text{ST}}(x) = \frac{1}{2}\sum_{i=1}^{d} \left(x_i^4 - 16x_i^2 + 5x_i\right).$$

To introduce more challenging and realistic scenarios, we further consider the following two variants of these base functions: **Heavy-tailed (HT) variant:** We add random noise to the function outputs, where the noise is sampled from a Weibull distribution. **Non-stationary (NS) variant:** We modulate each base function by a spatially-varying multiplicative factor:

$$f_{\text{NS}}(x) = (1 + \alpha \sin(x)e^x) \cdot f(x),$$

where $\alpha > 0$ controls the magnitude of non-stationarity. This formulation introduces location-dependent amplitude variations, breaking the stationarity assumption often made in kernel-based methods.

### C.2.2 Real-world Benchmark Details

**Portfolio** [Cakmak et al., 2020] We tune three hyperparameters of a trading strategy: risk aversion $[0.1, 1000]$, trade aversion $[5, 8]$, and holding cost multiplier $[0.1, 100]$. The environment includes two random variables: bid-ask spread $\sim \mathcal{U}[10^{-4}, 10^{-2}]$ and borrowing cost $\sim \mathcal{U}[10^{-4}, 10^{-3}]$. The objective is to maximize the $\text{VaR}_{0.8}$ of the average daily return over four years. To reduce cost, we build a GP surrogate model using 3000 evaluations selected via Sobol sampling, as in Cakmak et al. [2020].

**HPOBench** [Eggensperger et al., 2021] provides standard hyperparameter tuning tasks for ML models, and we use the MLP task in the HPOBench in our experiment.

**NASBench201** [Dong and Yang, 2020] is a neural network search task on CIFAR-100. Each architecture is represented by a 4-node directed acyclic graph with operations selected from a fixed set of 5 choices (e.g., conv, skip, zero). The total search space includes 15,625 architectures. Each architecture has been trained multiple times with fixed hyperparameters and its validation/test accuracy and training cost are pre-recorded. For our experiments, we query the test accuracy of a selected architecture based on its index as the ground-truth objective.

### C.2.3  Prompt Optimization Experiment Details

We conduct prefix prompt optimization experiments on seven language understanding datasets. All experiments are performed using the Alpaca-7B model [Taori et al., 2023], which remains frozen throughout. The decision variable is a continuous prefix embedding prepended to each input; its quality is evaluated based on downstream task performance (e.g., accuracy). Due to the high dimensionality of the prompt embedding space, we apply Uniform Manifold Approximation and Projection (UMAP) [McInnes et al., 2018] to project the original space into a lower-dimensional latent space.

- **SST-2** and **SST-5** are sentiment classification tasks based on the Stanford Sentiment Treebank [Socher et al., 2013]. SST-2 is a binary classification task where each sentence is labeled as either positive or negative. SST-5 is a more challenging 5-class variant with labels: very negative, negative, neutral, positive, and very positive. Both tasks are derived from movie review sentences and are standard benchmarks for evaluating natural language understanding.

- **MR** [Pang and Lee, 2005] is a sentiment classification dataset composed of full movie reviews with hidden star ratings. The authors constructed both three-class (negative, neutral, positive) and four-class versions of the dataset by removing explicit rating indicators and grouping documents by rating level.

- **CR** [Hu and Liu, 2004] is a sentiment classification dataset constructed from customer reviews of various electronic products collected from Amazon and CNET. Each review is labeled as either positive or negative at the sentence level.

- **AG's News** [Zhang et al., 2015] is a topic classification dataset consisting of news articles categorized into 4 classes: World, Sports, Business, and Science/Technology. Each sample includes the article's title and a short description. The dataset contains 120,000 training samples and 7,600 test samples, and is commonly used to benchmark text classification models. We treat this as a 4-class supervised learning task using the full text (title + description) as input.

- **TREC** [Voorhees and Tice, 2000] is a question classification dataset designed for evaluating systems that categorize natural language questions into broad types. The standard setup includes 6 coarse-grained classes (e.g., Person, Location, Numeric) and 50 fine-grained subclasses. We use the 6-way classification task with 5,452 training and 500 test examples.

- **Subj** [Pang and Lee, 2004] is a sentence-level subjectivity classification dataset consisting of 10,000 sentences: 5,000 subjective snippets from movie review sites and 5,000 objective sentences from plot summaries on IMDb. Each sentence is labeled as either subjective or objective.

| | GP-TS | GP-UCB | GP-EI | GP-KG | $\infty$-GP-TS |
|---|---|---|---|---|---|
| Time(second) | 2773 | 2649 | 3017 | 3388 | 2813 |

Table 2: Computation time (s) for five GP-based algorithms.

