# OpenReview forum: "Exploring and Exploiting Model Uncertainty in Bayesian Optimization"
_NeurIPS.cc/2025/Conference — NeurIPS 2025 poster_

### Official Review · Reviewer_44us · 2025-06-30

**Clarity:** 1
**Significance:** 3
**Originality:** 2
**Rating:** 3
**Confidence:** 2

**Summary:**

This paper considers Bayesian optimization where there is not only uncertainty in the reward model parameters but also in the reward model itself. They propose a new GP-based BO method to facilitate Bayesian optimization under this relaxed modeling assumption. The authors prove theoretical approximation guarantees and empirical results on synthetic experiments and LLM experiments.

**Questions:**

1. Could the authors elaborate more on the extension to high-dimensional BO problems, which were not included in the empirical valuation?
2. Could the authors describe in more detail how their setting differs from mis-specified BO?
3. Does equation (5) just expand the model class, and hence, allow one to avoid specifying a specific GP prior? To what extent is this broader class able to capture arbitrary reward models? (Apologies, I am not very familiar with SDP in this context.)

**Ethical Concerns:**

["NO or VERY MINOR ethics concerns only"]

**Final Justification:**

I thank the reviewers, authors, and ACs for their time. I will generally maintain my score of 3 due to remaining unconvinced about the technical novelty as raised in my initial review and my lack of familiarity with topic area. My main concern is
- I'm not sure how interesting / novel it is to extend existing frameworks to incorporate these new reward models, possibly because I am not a Bayesian optimization researcher.
- There is an absence of certain theoretical guarantees such as the regret bounds in the original manuscript. Although the authors claim it is easy to derive, I feel like I don't understand the main theoretical contribution still
I defer to other reviewers and the AC, as I am not an expert in this field.

**Limitations:**

Yes

**Quality:**

2

**Strengths And Weaknesses:**

Strengths
- the paper seems to tackle an important problem, which in my view is well-motivated by the discussion of model uncertainty in the introduction
- the paper includes both theoretical and empirical results, and the theoretical results match the minimax optimal rate, which is a strength
- it seems that the problem of model uncertainty has not directly been studied in the research previously

Weaknesses
- as someone who does not have a Bayesian optimization background (but still an ml/optimization theory background), I found the paper quite difficult to follow; this could be because I am not familiar with the landscape of the literature or because the writing lacks clarity. to be more specific, it was a bit difficult for me to understand how this paper sits in the related work, and I think that section in particular could be explained in a way that better explains the topic to a nonexpert in this specific subfield.
- on many of the experiments, it seems that confidence bars between the new method overlap.
- it is not clear to me how much the extension to capture model uncertainty constitutes as a

---

> ### Author Rebuttal · Authors · 2025-07-30
>
> We thank Reviewer 44us for the valuable feedback. We have addressed all concerns.
>
> >* [W1] Clarify the paper’s positioning in related work for non-BO experts.
>
> Thanks for the suggestion. To make the paper more accessible to non-experts, we will revise Section 2 to clearly highlight the two key components of BO, namely the acquisition policy and the surrogate model, and clarify that our contribution lies in proposing a new, more powerful surrogate. We will also revise the second paragraph of the Introduction and the paragraph of Related Work to better position our $\infty$-GP model as a replacement and enhancement of the standard GP, which serves as a fundamental backbone for BO and many uncertainty quantification tasks.
>
> **Here is a brief overview of our contribution. GP is the foundational surrogate model in BO. We propose an enhancement of this backbone**: the $\infty$-GP model, which defines an infinite mixture of GPs while remaining tractable in practice. This enables modeling a much richer class of complex reward distributions. The intuition is similar to the universal approximation theorem: one can approximate a complex function using an infinite sum of simple base functions. At the same time, $\infty$-GP inherits the nice properties of standard GPs, such as closed-form updates and extensibility.
>
> >* [W2] Concern regarding overlapping confidence bars:
>
> While our method’s confidence intervals partially overlap with those of some baselines, the overlap is minor, and our method consistently achieves significantly better mean performance. BTW, we believe this overlapping phenomenon is fairly common and expected in BO (e.g., as also observed in [1]). One reason for the overlap is that BO algorithms typically encourage exploration, which introduces more randomness and can lead to wider confidence intervals.
>
> [1] Xu, Zhitong, et al. "Standard Gaussian Process is All You Need for High-Dimensional Bayesian Optimization." The Thirteenth International Conference on Learning Representations.
>
> >* [Q1] Question regarding the extension to high-dimensional BO:
>
> Since the $\infty$-GP model is defined as a mixture over infinitely many GPs, each layer corresponds to a standard GP. By modifying the base GP in each layer, **our framework can seamlessly incorporate any existing GP extensions, including those designed for high-dimensional settings**. For example, sparse GPs, Gaussian Markov random fields, additive GPs, GP with low-dimensional subspace embeddings, and high-dimensional BO methods utilizing deep learning.
>
> Indeed, **our experiment on the prompt optimization task already provides some empirical evidence on how the $\infty$-GP framework can be extended to high-dimensional settings**. In this task, we use UMAP for dimensionality reduction, which can be viewed as a low-dimensional embedding-based BO method, and observe strong empirical performance under this setting.
>
> >* [Q2] Clarify how the proposed setting differs from mis-specified BO:
>
> Thanks for the question. Existing mis-specified BO methods typically handle limited types of model mismatch, such as incorrect smoothness assumptions, within a fixed model class like GPs. In contrast, our $\infty$-GP model places a prior over the entire space of reward distributions, allowing us to capture model uncertainty *across* different function classes and providing much greater flexibility.
>
>
> >* [Q3] Clarify whether Eq. (5) simply expands the model class and how expressive it is:
>
> Thanks for the question. Your understanding of Eq.(5) is essentially correct, but more precisely, **we are not simply expanding the model class; instead, we let the reward model (which is a distribution) itself be random and place a prior over the space of reward distributions**. Regarding “to what extent is this broader class able to capture arbitrary reward models,” the discussion following Assumption 1 rigorously characterizes the class of reward distributions that can be approximated by our $\infty$-GP model, and shows that it is significantly broader than that of standard GP model. We also note that our paper does not require the reader to have prior knowledge of SDP, as we clearly define the SDP and provide intuitive explanations to aid understanding.
>
> >[W3]
>
> The comment in Weakness 3 appears incomplete. We would appreciate it if the reviewer could kindly clarify or complete the sentence, so we can provide a proper and constructive response.

---

> > ### Author Response · Authors · 2025-08-05
> > **kind reminder**
> >
> > Dear Reviewer 44us
> >
> > Thank you once again for your valuable comments on our submission. As the discussion phase is approaching its end, we would like to kindly confirm whether we have sufficiently addressed all of your concerns (or at least part of them). Should there be any remaining questions or areas requiring further clarification, please do not hesitate to let us know. If you are satisfied with our responses, we would greatly appreciate your consideration in adjusting the evaluation scores accordingly.
> >
> > We sincerely look forward to your feedback.

---

> > > ### Comment · Reviewer_44us · 2025-08-05
> > > **Response to rebuttal**
> > >
> > > I appreciate the authors for getting back to me! I apologize that W3 was cut off, I think it was an error in copying my response from my text editor.
> > >
> > > I meant to ask:
> > >
> > > "It is not clear to me the degree to which the extension to capture model uncertainty constitutes a novel contribution as opposed to an incremental update on prior works in GP."
> > >
> > > The response clarifying the role of equation (5) was helpful as well.

---

> > > > ### Author Response · Authors · 2025-08-05
> > > >
> > > > Many thanks for your helpful clarification! As you noted, our explanation of Eq.(5) already addresses part of your concern in W3, as it demonstrates how incorporating model uncertainty enables us to go beyond the restrictive assumptions of standard GPs and converge to a much broader class of reward distributions.
> > > >
> > > > Additionally, we would like to emphasize that **enabling model uncertainty quantification (UQ) is itself a significant contribution to the UQ literature**. UQ is a central topic in reliable ML and AI, and one of the key reasons for the popularity of GP is its ability to provide UQ. However, traditional GPs only quantify uncertainty within a fixed model class, while ignoring the **uncertainty about whether the model class itself is correct**. Our $\infty$-GP fills this important gap.
> > > >
> > > > If our response has addressed your concern, we would be sincerely grateful if you would consider updating your score.

---

> > > > ### Author Response · Authors · 2025-08-08
> > > > **Kind remainder**
> > > >
> > > > Thank you again for your time and engagement!
> > > >
> > > > As the rebuttal phase is nearing its end and the other three reviewers have largely completed their discussions, we would like to kindly invite you back to this discussion.
> > > >
> > > > We believe we have addressed all the questions you previously raised (including W3). If there are still any unresolved points, we would be more than happy to continue the discussion and provide additional clarifications to further strengthen the paper. If our response has addressed your concerns, we would be sincerely grateful if you could consider updating your score accordingly.

---

> > > > > ### Comment · Reviewer_44us · 2025-08-09
> > > > > **Thanks**
> > > > >
> > > > > Thank you for your response, I appreciate the clarifications. I will consider the response and update my score after the AC-reviewer discussions.

---

> > > > > > ### Author Response · Authors · 2025-08-09
> > > > > > **Thanks**
> > > > > >
> > > > > > Thanks for your time and comments! We sincerely appreciate the efforts of both you and the AC in helping to improve our paper during the discussion phase.

---

### Official Review · Reviewer_JUCH · 2025-06-30

**Clarity:** 2
**Significance:** 2
**Originality:** 3
**Rating:** 4
**Confidence:** 4

**Summary:**

- Traditional Gaussian Process (GP) surrogate-based Bayesian Optimisation (BO) struggles with non-stationary, multi-modal, and heavy-tailed objective function distributions $f(x)$ due to strong structural assumptions. Most of the standard GPs capture value uncertainty but not model uncertainty, leading to poor performance when the underlying Gaussian structural assumptions are violated.

- This paper introduces a $\infty-$Gaussian Process ($\infty-$GP), a novel surrogate that constructs the reward function or the objective function as an infinite mixture of GP surfaces via a spatial Dirichlet Process (SDP). The proposed $\infty-$GP is paired with the Thompson Sampling (TS) strategy that samples a surrogate function from the posterior and maximises it to select the next candidate for function evaluation. Additionally, they also balance between exploring the unknown search surfaces and exploiting the known search surfaces to find the optima.

- The empirical results provided outperform GP-based BO methods on both synthetic and real-world tasks, especially in non-stationary and heavy-tailed noise settings. Authors have also demonstrated the applicability of their proposed method in complex tasks like LLM prompt tuning.

**Questions:**

- How is the scalar precision parameter $\nu$ determined? While its prior distribution is mentioned as a Gamma distribution in Appendix B.1 (Equation 27), the specific values or settings for the hyperparameters $a_\nu$ and $b_\nu$ are not clearly stated. Clarification on these parameters would be helpful.

- How sensitive are the hyperparameters priors used in the posterior inference procedure (Gibbs Sampling)?

- Is there a specific statistic/metric that quantifies the reuse of latent surfaces across iterations to support the claims about exploitation behaviour of the proposed approach (discussed in Lines 221–223)?

- What is the theoretical order of the regret for the proposed method? Does it maintain sub-linear regret, similar to standard Thompson Sampling-based BO? If so, what are the key theoretical differences due to operating in the distributional space, and would they affect regret bound?

- How does the proposed approach compare with existing BO methods that exploit additive structures of the objective function [1]? Given that the $\infty-$GP uses multiple latent GP surfaces, does it implicitly capture the additive nature of the objective function, or are there fundamental differences?

- While the authors address non-stationarity via input warping, several BO methods use explicitly non-stationary kernels or model heteroskedastic noise directly. Why were such approaches not included in the comparisons? A direct evaluation against BO methods with non-stationary kernels would provide a more thorough benchmark and further strengthen the benefits of the proposed approach.



 - **Minor Questions**
 	- What does the y-axis represent in Figure 2? Would normalisation or standardisation of the reward values impact the performance or stability of the proposed method?
   - What is the number of input dimensions used in the synthetic function benchmarks? Including this detail would help assess the scalability and complexity of the evaluation tasks.

**Ethical Concerns:**

["NO or VERY MINOR ethics concerns only"]

**Final Justification:**

After reading the author's rebuttal and engaging in a discussion with them, I am keeping my final score as "Borderline Accept."

**Limitations:**

Although the authors acknowledge the issue of scalability of the proposed approach to high dimensions, there are a few other important limitations that are not mentioned in the manuscript.

- The method relies on Gibbs sampling for posterior inference over multiple GP surfaces, which is computationally intensive, especially for large datasets.

- The performance of the $\infty-$GP model is highly dependent on the careful selection of hyperparameter priors and these priors are often domain-specific and may not generalise well across different tasks.

- The paper does not provide formal regret bounds for the proposed BO strategy, thereby limiting its theoretical interpretability and comparability to standard BO methods (backed by known sub-linear regret guarantees).

**Paper Formatting Concerns:**

- The authors have not followed the NeurIPS 2025 formatting guidelines regarding section headings. Paragraphs are used in place of subheadings.

- Table 1 does not follow the standard NeurIPS format, as the caption is placed above the table rather than below it, as mentioned in the NeurIPS 2025 template.

**Quality:**

3

**Strengths And Weaknesses:**

**Strengths**

- In this paper authors bring in the notion of _Model Uncertainty_ aspects of the BO Process, in contrast to the _value uncertainty_ generally considered in the Bayesian optimisation literature.

- Inspired by the prior literature on infinite mixture of GPs, authors have devised a novel approach $\infty-$GP, that models the objective function as an infinite mixture of GP surfaces via a spatial Dirichlet Process (SDP).

- The proposed $\infty-$GP accounts for uncertainty in the choice of surrogate model itself (model uncertainty), not just the output values (value uncertainty), by allowing each input location to probabilistically mix over infinite GP surfaces.

- Authors have combined $\infty-$GP surrogate model with Thompson Sampling (TS) acquisition function to effectively balance between the exploration of the unknown regions and exploitation of the known regions/solutions.

- In contrast to exploration-exploitation in the traditional TS-based BO methods, the proposed approach discusses the exploration-exploitation tradeoff in the distributional space wherein the exploitation refers to probabilistic selection of the existing surfaces and exploration refers to introducing a new GP surface for realising the candidate points.


**Weaknesses**

- The authors have performed posterior inference using a Gibbs sampling strategy under a fully Bayesian model. In general, Gibbs sampling over infinite (or many) GP surfaces and posterior inference can be computationally intensive, especially for large datasets. Authors have also mentioned that the number of mixture components is set to a finite value $L$ (Line 571 in B.2 of Appendix), contradicting the motivation to use an infinite mixture of GP surfaces. Further, the performance may degrade if $L$ is not chosen appropriately.

- The success of the proposed sampling strategy and thus the overall BO procedure relies heavily on the prior specification of the hyperparameters (Equations 24 to 27 in Appendix B.1), including kernel width, noise variance, and the Dirichlet process concentration parameter. These priors are often highly domain-specific and task-sensitive, which may limit the method’s generalizability without some tuning efforts.

- In the ablation results, the authors concluded that the reward space can be effectively modelled as a mixture of multiple latent GP surfaces. However, the paper does not discuss the existing BO literature that leverages the additive structure of the objective function. Incorporating a discussion of such approaches could provide some context and help understand the advantages of the proposed method.

- Although the empirical analyses demonstrate improved performance over classical GP-based BO methods, it does not include comparisons against state-of-the-art approaches specifically designed to address non-stationarity [1,2] or exploit additive structure [3] in the objective function space, including such baselines would provide a comprehensive evaluation of the proposed method’s strengths and limitations.

- In the manuscript, the authors do not discuss the implications of the sampling strategy on the regret bounds of the overall Bayesian optimisation process. Discussing the regret bounds of the proposed approach could be highly beneficial to compare with other state-of-the-art methods that generally discuss the space/time complexities and the regret bounds.

**References**

[1] Ruben Martinez-Cantin. Funneled Bayesian optimisation for design, tuning and control of autonomous systems. IEEE Transactions on Cybernetics, 49(4):1489– 1500, 2019.

[2] Remes, Sami, Markus Heinonen, and Samuel Kaski. "Non-stationary spectral kernels." Advances in neural information processing systems 30 (2017).

[3] Gardner, Jacob, Chuan Guo, Kilian Weinberger, Roman Garnett, and Roger Grosse. "Discovering and exploiting additive structure for Bayesian optimisation." In Artificial Intelligence and Statistics, pp. 1311-1319. PMLR, 2017.

---

> ### Author Rebuttal · Authors · 2025-07-30
>
> We sincerely thank the reviewers for their thoughtful comments. We have carefully addressed all the questions and concerns raised.
> >* [W1] Regarding the computational complexity of Gibbs sampling and the choice of truncation number $L$
>
> First, we use a truncation technique in Gibbs sampling, i.e. setting the maximum number of mixture component to fixed $L$. By doing so, the complexity of truncated Gibbs sampling is $\mathcal{O}(n^3)$, which is exactly **equal to traditional GP** (for more details, please refer to our response to W5&Q3).
>
> Second, despite using a finite truncation, we must note that **we are not "contradicting the motivation to use an infinite mixture of GP surfaces", and our model is still theoretically an infinite mixture of GPs**. This is because, as shown in Eq. (10), when we actually perform posterior prediction using the $\infty$-GP, we do not use infinitely many surfaces, but rather $K_n+1$ surfaces. The truncation level $L$ essentially imposes an upper bound on $K_n$, and in our experiments, we set $L=8$. In practice, this upper bound is rarely reached, and thus the performance is largely insensitive to the choice of $L$. This observation is supported both theoretically and empirically. **Theoretically**, as shown in Equation (2.6) of [1], the number of active surfaces $K_n$ in a Dirichlet Process grows very slowly with $n$, at a rate of $\log n$. **Empirically**, the table below reports the best found values (optimal = 0) for the Ackley-HT-NS task under different values of $L$. The performance is not sensitive to $L$. This is because, by the 100th iteration, we observe that only 2–3 surfaces have non-negligible weights in the posterior distribution. The remaining layers are essentially unused, so When $L$ is 6 or greater, changing it has little impact on the result.
>
>
> | $L$   | 6     | 7     | 8     | 9     | 10    |
> |-------|-------|-------|-------|-------|-------|
> | Best Found Value | -0.10 | -0.09 | -0.08 | -0.11 | -0.09 |
>
> [1] Bayesian nonparametrics[M]. Cambridge University Press, 2010.
>
> >* [W2 & Q1 & Q2] Prior specification and sensitivity to hyperparameter choices.
>
> Thanks for the questions. The prior specification involves two parts: (i) choosing the prior distribution families, and (ii) setting the hyperparameters for those priors. We explain the two parts one by one below.
>
> (i) We use conjugate priors for $\sigma^2$, $\tau$ and $\nu$ to allow efficient Gibbs sampling with analytical posterior updates. These choices are quite standard and make the inference much easier in practice. For $\phi$, we follow the recommendation from [2] and use a uniform prior, which is shown in their paper to work well and be numerically stable.
>
>
> (ii) The prior hyperparameters include $a_\nu, b_\nu, a_\tau, b_\tau, a_\sigma, b_\sigma, b_\phi$. We did not include the precision prior hyperparameter $(a_\nu, b_\nu)$ in the appendix, but in our experiments, we simply set it to $(1,1)$. The settings for the other hyperparameters are already listed. We note that, **except for $b_\phi$, the results are not sensitive to these prior hyperparameters**. Please refer to our response to **[W1 & Q2] of Reviewer v6ga** for a detailed sensitivity analysis of the prior hyperparameters, where we provide both empirical results and theoretical justifications. We also provide guidance on how to choose the $b_\phi$.
>
> [2] Gelfand A E, Kottas A, MacEachern S N. Bayesian nonparametric spatial modeling with Dirichlet process mixing[J]. Journal of the American Statistical Association, 2005, 100(471): 1021-1035.
>
> >* [W3 & W4 & Q5 & Q6] Comparison to additive GP and non-stationary kernel baselines
>
> Thank you for these insightful suggestions. We will include relevant comparisons to BO methods that exploit additive structure and use explicitly non-stationary kernels  in the revised version. We would like to clarify that **the proposed $\infty$-GP is fundamentally different from additive GP models**. Additive BO are designed for high-dimensional problems by partitioning the input dimensions and assigning a single GP surrogate to each subset of dimensions. The resulting model is still a single GP but with an additive kernel. In contrast, our $\infty$-GP framework models the entire input space globally as a mixture of infinitely many GP surfaces. The resulting model is no longer a GP, and can represent reward models that lie far beyond the GP function class. By the way, **additive BO can be seamlessly integrated into the $\infty$-GP framework** by replacing the standard GP in each surface with an additive GP.
>
> As recommended, we will include the following experimental results comparing our method with TS using additive GP in [4] and non-stationary spectral kernel GPs in [3]. Both alternatives underperform our method. This is because both baselines still operate within the single-GP framework, which is limited in its ability to model complex rewards.
> |  | Portfolio | HPOBench | NASBench |
> |-|-|-|-|
> | Additive-TS | 9.0 | 67.3 | 65.0 |
> | Nonstationary-TS | 10.4 | 65.2 | 68.5 |
> | $\infty$-GP-TS | 11.4 | 68.7 | 70.5 |
>
> [3] Remes, Sami, Markus Heinonen, and Samuel Kaski. "Non-stationary spectral kernels." Advances in neural information processing systems 30 (2017).
>
> [4] Gardner, Jacob, Chuan Guo, Kilian Weinberger, Roman Garnett, and Roger Grosse. "Discovering and exploiting additive structure for Bayesian optimisation." In Artificial Intelligence and Statistics, pp. 1311-1319. PMLR, 2017.
>
>
> >* [W5 & Q4] Regarding regret analysis and time complexity analysis
>
>
> Thanks for your helpful comment. We will include regret and time complexity analysis in the updated version of the paper.
>
> * Regret: **The $\infty$-GP-TS indeed maintains sub-linear regret.** In fact, we only need one additional step from our established Theorem 1 to derive a regret bound. The instantaneous regret at step $n$ satisfies: $r_n\leq\int |\mu^\ast(x)||\pi_{f_n}(x)-\pi_{f^\ast}(x)|dx\leq2\mu^\ast(x_\ast)\delta(\pi_{f_n},\pi_{f^\ast})\leq2\mu^\ast(x_\ast)\delta(f_n,f^\ast)$, where $\delta$ is total variation distance (TVD), $\pi_{f_n}$ and $\pi_{f^\ast}$ is the TS policy distribution induced by surrogate posterior $f_n$ (defined in Eq.(8)) and true reward $f^\ast$, respectively.
>  The last inequality follows from the *data processing inequality*. The convergence $\delta(f_n,f^\ast)\to 0$ has already been established in Theorem 1 (noting that L1 norm= 2TVD). Therefore, we obtain $r_n\to 0$, implying sublinear cumulative regret. Moreover, the concrete regret rate can be made explicit by leveraging the posterior contraction result in Theorem 1.
> Regarding the question about "the key theoretical differences due to operating in the distributional space," we would like to clarify the following: First, our model indeed defines a prior over the distributional space (i.e., $G$ is drawn from an SDP). However, as shown in Eq. (13), we marginalize out the random distribution $G$ over the SDP when computing the posterior predictive distribution. As such, the actual $\infty$-GP-TS algorithm does not explicitly operate over the distributional space. Second, the theoretical difference is as follows: Traditional GP-TS analyzes regret by studying how the posterior predictive mean concentrates around the true function. In contrast, in $\infty$-GP-TS, as shown in Theorem 1 and the above analysis, we bound the regret in terms of the contraction rate of the posterior over the distributional space toward the true reward distribution.
>
>
>
> * Time Complexity: By using a truncation technique (see Algorithm 2), **the complexity of our method is $\mathcal{O}(n^3)$, which is equal to that of strandard GP-based BO.** In short，the complexity is governed by two main components. (i) The computational complexity of Kriging, which is mainly govern by computing the inverse $\Sigma_0(x_{1:n}, x_{1:n})^{-1}$, incurring $\mathcal{O}(n^3)$ cost.  However, the $\infty$-GP model enjoys the advantage that all surfaces share the same $\Sigma_0(x_{1:n}, x_{1:n})^{-1}$, allowing this expensive matrix inversion to be computed only once. (ii) The complexity of Gibbs sampling. By truncating the maximum number of surface to $L$, one can sample the $\xi_l(x_{1:n})$ jointly and thus the complexity is $\mathcal{O}(n^3)$. In summary, the total complexity of $\infty$-GP is $\mathcal{O}(n^3)$. The time complexity analysis is supported by the following wall-clock time results (in seconds).
>
> |    Objective    | GP-TS | GP-UCB | GP-EI | GP-KG | $\infty$-GP-TS |
> |----------------|-------|--------|-------|--------|----------------|
> | Ackley-HT      | 2773  | 2649   | 3017  | 3388   | 2813           |
>
>
> >* [Q3] Question about metric that quantifies the reuse of latent surfaces
>
>  Thank you for this insightful question. The reuse of latent surfaces can indeed be quantified with the following quantities. First, $n_j$ quantifies how many times the surface $j$ has been reused in past iterations. Second, the posterior of $\nu$ characterizes the decision maker’s model uncertainty and governs the trade-off between introducing new surfaces (exploration) versus reusing existing ones (exploitation). Smaller values of $\nu$ encourage more reuse. Third, the ratio $\frac{{n}_j}{\nu + n}$ quantifies the probability of reusing surface $j$ in the upcoming new iteration.
>
> >* [Q7] Minor questions
>
> The y-axis in Figure 2 denote the original value of the test function. In our experiments, all methods—including both the baseline BO approaches and our $\infty$-GP-TS use sklearn.MinMaxScaler to linearly map observed rewards into the fixed interval $[-5,5]$ before fitting the surrogate. It is a common practice for BO to ensure numerical stability. And all methods use the same affine transform to ensure fair comparisons. After fitting the surrogate, we use the inverse transform to show the final result in the paper. The dimensions for the synthetic function is 20.

---

> > ### Author Response · Authors · 2025-08-05
> > **Kind reminder**
> >
> > Dear Reviewer JUCH
> >
> > Thank you once again for your valuable comments on our submission. As the discussion phase is approaching its end, we would like to kindly confirm whether we have sufficiently addressed all of your concerns (or at least part of them). Should there be any remaining questions or areas requiring further clarification, please do not hesitate to let us know. If you are satisfied with our responses, we would greatly appreciate your consideration in adjusting the evaluation scores accordingly.
> >
> > We sincerely look forward to your feedback.

---

> ### Comment · Reviewer_JUCH · 2025-08-05
> **Reply to Author Rebuttal**
>
> I have read the rebuttal and appreciate the authors’ efforts in addressing all my concerns. The additional insights, particularly the discussion on regret bounds and time complexity, strengthen the paper and could enhance its acceptance within the community. While the authors have provided their rationale for choosing the prior distribution families, I remain somewhat unconvinced about the practical feasibility of this approach. Nevertheless, I will retain my score as "Borderline Accept."

---

### Official Review · Reviewer_BoMR · 2025-07-03

**Clarity:** 2
**Significance:** 3
**Originality:** 3
**Rating:** 4
**Confidence:** 3

**Summary:**

This paper introduces a novel surrogate model for Bayesian Optimization (BO), the infinity-Gaussian Process ($\infty$-GP), designed to address challenges where the reward distribution is complex, non-stationary, or heavy-tailed. The $\infty$-GP is formulated as a sequential spatial Dirichlet Process mixture with a Gaussian Process (GP) baseline. This approach allows the model to quantify not just value uncertainty but also *model uncertainty*—uncertainty about the underlying distributional form of the reward. The authors provide a theoretical analysis demonstrating that the $\infty$-GP posterior can approximate a broad class of reward distributions with a near-minimax optimal posterior contraction rate. Empirically, the proposed $\infty$-GP combined with Thompson Sampling ($\infty$-GP-TS) is shown to outperform standard GP-based BO methods on various challenging synthetic and real-world benchmarks, including a prompt optimization task.

**Questions:**

1.  What is the conceptual relationship between the proposed $\infty$-GP model and the **spectral mixture kernel**? Both are designed to model complex, non-stationary functions. Does $\infty$-GP offer any distinct advantages over using a highly flexible kernel within a standard GP framework?

2.  Could the authors further elaborate on the statement in lines 143-145 regarding the role of the noise term $\epsilon_i$? The authors state it is introduced for "modeling purposes to mix with the discrete-distributed $\xi(x)$ and produce a continuous reward distribution".

3.  Could you provide more specific details for the empirical studies to improve reproducibility? Specifically:
    * **HPOBench:** What is the specific task ID for the MLP experiment from the benchmark suite?
    * **Prompt Engineering:** How was the UMAP projection configured (e.g., number of components, neighbors)? Were other dimensionality reduction techniques considered?
    * **General:** How many MCMC iterations (B) and mixture components (L) were used in the Gibbs sampler (Algorithm 2) for the experiments?

**Ethical Concerns:**

["NO or VERY MINOR ethics concerns only"]

**Final Justification:**

Despite having issues connecting the currently presented results to the canonical cumulative regret bound, I believe the novel theoretical insight on the convergence of the simple regret, and the empirical results showing improvement on some well-known tasks qualify as acceptance.

**Limitations:**

Not discussed.

**Paper Formatting Concerns:**

No.

**Quality:**

3

**Strengths And Weaknesses:**

#### **Strengths**

- **Novelty and Significance:** The paper proposes a novel and principled approach ($\infty$-GP) to handle model misspecification in BO, which is a significant and practical problem. Standard GP-based methods often fail when their strong modeling assumptions are violated, and the $\infty$-GP offers a flexible alternative by creating an infinite mixture of GP surfaces.
- **Theoretical Justification:** A solid theoretical contribution supports the work. Theorem 1 establishes posterior convergence guarantees for the $\infty$-GP model, demonstrating its ability to approximate a broad class of reward distributions at a near-minimax-optimal rate. This provides confidence in the model's robustness, extending beyond its empirical performance.
- **Empirical Validation:** The experimental evaluation is comprehensive, covering synthetic benchmarks with non-stationary and heavy-tailed properties, real-world tasks like portfolio optimization and neural architecture search, and a modern application in LLM prompt optimization. The results consistently show the superiority of $\infty$-GP-TS over standard GP baselines in these challenging settings.


**Weaknesses**

1.  **Limited Literature Review on Model Uncertainty:** The concept of "model uncertainty" is framed as a key motivator, yet the literature review on this topic feels incomplete. This issue has been previously addressed under the umbrella of hyperparameter learning for surrogate models. A relevant and missing work is "Self-Correcting Bayesian Optimization through Bayesian Active Learning" by Hvarfner et al. (2024), which also actively manages model uncertainty. The current related work section could be strengthened by positioning the $\infty$-GP in the context of these alternative approaches.

2.  **Missing State-of-the-Art Baselines:** The paper compares $\infty$-GP-TS against four standard academic baselines: GP-TS, GP-UCB, GP-EI, and GP-KG. While these are foundational, the comparison lacks more recent, powerful, and industrially relevant baselines. For instance, HEBO, which explicitly handles heteroscedastic and non-stationary rewards via input and output warping, has demonstrated state-of-the-art performance. The recent Vizier report (2024) also highlights robust methods for these scenarios. Including a comparison to at least one such advanced baseline would provide a more convincing measure of the proposed method's practical utility.

3.  **Lack of Regret Analysis:** While the paper provides a posterior convergence guarantee (Theorem 1), it does not offer a theoretical analysis of the cumulative regret for the proposed $\infty$-GP-TS algorithm. Regret bounds are a standard way to characterize the sample efficiency of a BO algorithm. Without this analysis, the theoretical understanding of the algorithm's exploration-exploitation trade-off remains incomplete.

4. **Insufficient Experimental Details:** Although the empirical studies are broad, they lack key details necessary for full reproducibility.
    * For **HPOBench**, the paper states it uses the "MLP task" but does not specify the task ID or the number of training epochs, making it difficult to replicate the results.
    * The **prompt optimization** section is missing comparisons to domain-specific BO baselines. The setup could also be clarified further regarding the construction of the UMAP search space.
    * The hyperparameter choices for each experiment are not explicitly discussed.


Reference:
- Hvarfner, Carl, Erik Hellsten, Frank Hutter, and Luigi Nardi. "Self-correcting bayesian optimization through bayesian active learning." Advances in Neural Information Processing Systems 36 (2023): 79173-79199.

- Cowen-Rivers, Alexander I., Wenlong Lyu, Rasul Tutunov, Zhi Wang, Antoine Grosnit, Ryan Rhys Griffiths, Alexandre Max Maraval et al. "Hebo: Pushing the limits of sample-efficient hyper-parameter optimisation." Journal of Artificial Intelligence Research 74 (2022): 1269-1349.

- Song, Xingyou, Qiuyi Zhang, Chansoo Lee, Emily Fertig, Tzu-Kuo Huang, Lior Belenki, Greg Kochanski et al. "The vizier gaussian process bandit algorithm." arXiv preprint arXiv:2408.11527 (2024).

---

> ### Author Rebuttal · Authors · 2025-07-30
>
> We sincerely thank the reviewers for their thoughtful comments and constructive feedback. We have carefully addressed all the questions and concerns raised, and have made corresponding revisions or clarifications as detailed below.
>
> >* [W1]:Missing discussion on existing literature on model uncertainty in hyperparameter learning
>
> Thank you. We will revise the last paragraph of the “Related Work” section to explicitly contrast our $\infty$-GP framework with prior work on model uncertainty in hyperparameter learning and related areas, including Hvarfner et al. (2024) and other relevant studies such as [1,2]. Specifically, we will clarify that **those approaches manage model uncertainty within a fixed model class (e.g., GPs with uncertain kernel design), whereas our $\infty$-GP addresses uncertainty *over* model classes** by placing a prior over the distributional space of reward models, offering greater modeling flexibility.
>
> [1] Malkomes G, Garnett R. Automating Bayesian optimization with Bayesian optimization[J]. Advances in Neural Information Processing Systems, 2018, 31.
>
> [2] Malkomes G, Schaff C, Garnett R. Bayesian optimization for automated model selection[J]. Advances in neural information processing systems, 2016, 29.
>
> >* [W2] Missing comparison with state-of-the-art industrially relevant methods such as HEBO.
>
> Thanks for your helpful comments. In response, **we have included HEBO as an additional baseline in our updated experiments** (see the table below, which reports the best-found performance after 100 iterations). Our method consistently outperforms HEBO in Portfolio task, HPOBench and NASBench.
>
> | method | Portfolio | HPOBench | NASBench |
> |-|-|-|-|
> | HEBO | 10.4 | 67.3 | 69.2 |
> | $\infty$-GP-TS | 11.4 | 68.7 | 70.5 |
>
> First, while HEBO incorporates input and output warping to address non-stationarity (in fact, in our experiments, all baselines for nonstationary tasks already adopt input warping techniques by default), it still relies on a single GP model as the surrogate. As such, the model class that a single GP can represent remains limited, and it cannot handle heavy-tailed noises. Our results show that warping alone is insufficient. Second, HEBO ensembles multiple base acquisition functions (e.g., EI, UCB) to improve search efficiency. However, as our results demonstrate, in highly complex reward landscapes involving model misspecification, the choice of a more expressive surrogate model (such as $\infty$-GP) is often more critical than acquisition efficiency. Importantly, the complex structure of our $\infty$-GP surrogate is learned directly from data, rather than hand-crafted through warping techniques.
>
> >* [W3] Lack of Regret Analysis
>
> Thanks for your helpful comment. We will include regret analysis in the updated version of the paper. In fact, we **only need one additional step** from our established Theorem 1 to derive a regret bound. The instantaneous regret at step $n$ satisfies: $$r_n\leq\int |\mu^\ast(x)||\pi_{f_n}(x)-\pi_{f^\ast}(x)|dx\leq2\mu^\ast(x_\ast)\delta(\pi_{f_n},\pi_{f^\ast})\leq2\mu^\ast(x_\ast)\delta(f_n,f^\ast),$$ where $\delta$ is total variation distance (TVD), $\pi_{f_n}$ and $\pi_{f^\ast}$ is the TS policy distribution induced by surrogate posterior $f_n=f(y(x_n)|\mathcal{H}_n)$ (defined in Eq.(8)) and true reward $f^\ast$, respectively. The last inequality follows from the *data processing inequality*. The convergence $\delta(f_n,f^\ast)\to 0$ has already been established in Theorem 1 (noting that L1 norm= 2TVD). **Therefore, we obtain $r_n\to 0$, implying so-called no-regret or sublinear cumulative regret.**
>
> Moreover, the concrete regret rate can be made explicit by leveraging the posterior contraction result in Theorem 1, requiring only a standard discretization argument (e.g., from [3]) to handle the continuous decision space.
>
> [3] Srinivas N, Krause A, Kakade S M, et al. Gaussian process optimization in the bandit setting: No regret and experimental design[J]. arXiv preprint arXiv:0912.3995, 2009.
>
> >* [W4&Q3] Regarding Experimental Details:
>
> **(1)HPObench details:** We would like to clarify that we used the "kc1" dataset for the MLP task, which is widely adopted in the AutoML and HPO benchmark community. The OpenML task ID for kc1 is 3917. This dataset consists of 2,109 training instances and 22 features (after preprocessing), and it originates from a real-world software defect prediction task.
>
> **(2)Prompt optimization details:** We apply UMAP to compress high-dimensional prompt embeddings into a 10-dimensional manifold-preserving latent space. The specific UMAP configuration is as follows: we set the number of components to 10 (i.e., n_components=10), and the number of neighbors to 15 (n_neighbors=15). We also used the 'cosine' distance metric, as it is more suitable for semantic embeddings than the default Euclidean metric. The minimum distance parameter was set to 0.0 to encourage tighter clusters in the projected space. Other dimensionality reduction techniques such as PCA and t-SNE were considered, but we chose UMAP because it preserves both local and global structure, is computationally efficient for large text collections, and allows flexible adjustment of clustering granularity via the n_neighbors parameter.
>
>
> Regarding the comment on “missing comparisons to domain-specific BO baselines,” to the best of our knowledge, there is no established class of BO algorithms specifically tailored for prompt optimization that fundamentally differs from general-purpose BO methods. Notable works, including [4] and [5], all rely on standard BO algorithms such as GP-UCB or GP-EI. Compared with our implementation, their main difference lies in the way they embed or reduce the dimensionality of the prompt space. To ensure a fair comparison focused on the BO algorithm itself rather than the embedding method, we adopt UMAP across all methods.
>
> **(3) Hyperparameter setting:** In our experiments, we set the number of MCMC iterations to B = 500 and the truncation number of mixture components to L = 8.
>
> [4] Chen L, Chen J, Goldstein T, et al. Instructzero: Efficient instruction optimization for black-box large language models[J]. arXiv preprint arXiv:2306.03082, 2023.
>
> [5] Sabbatella A, Ponti A, Giordani I, et al. Prompt optimization in large language models[J]. Mathematics, 2024, 12(6): 929.
>
> >* [Q1] Clarify difference between $\infty$-GP and spectral mixture kernel:
>
> The spectral mixture kernel (SMK) enhances the expressiveness of a **single GP** by constructing flexible kernels through spectral mixtures. However, these methods still assume the reward process lies within the GP family, thus limiting their modeling flexibility. In contrast, $\infty$-GP places a prior over the space of reward distributions, allowing us to model rewards that are not themselves GPs. This enables model uncertainty over function classes, rather than within a fixed class. **Compared to using a highly flexible kernel within a standard GP (e.g., SMK), the $\infty$-GP offers several distinct advantages**.
> * First, it provides greater modeling flexibility and accommodates not only non-stationary rewards, but also heavy-tailed, heteroskedastic, and other non-Gaussian noise structures, as supported by our theoretical guarantee in Theorem 1.
> * Second, **SMK can be directly incorporated into $\infty$-GP framework** by replacing the base kernel. Since $\infty$-GP is a mixture over infinitely many GPs, it is compatible with any GP extensions, including custom kernel designs and other architectural improvements.
> * Third, unlike standard GPs, $\infty$-GP quantifies not only value uncertainty but also model uncertainty.
>
> >* [Q2] question regarding the role of the noise term
>
> Let us further explain the role of the noise term $\epsilon$. As defined in Eq. (6), $\xi$ is discretely distributed (though with infinite support), so it cannot directly model a continuous reward distribution. By mixing $\xi$ with a Gaussian $\epsilon$, the resulting distribution of $y(x)=\xi(x)+\epsilon(x)$ becomes continuous. Here, the term $\epsilon$ does not mean that we impose a Gaussian assumption on the true measurement noise, as we directly model the overall distribution of the observation $y$ via Eq. (4), rather than modeling the expectation and measurement noise separately as in classic GP-based BO. Therefore, we say the term $\epsilon$ is introduced solely for “modeling purposes”, without real physical meaning.

---

> ### Author Response · Authors · 2025-08-05
> **Kind reminder**
>
> Dear Reviewer BoMR
>
> Thank you once again for your valuable comments on our submission. As the discussion phase is approaching its end, we would like to kindly confirm whether we have sufficiently addressed all of your concerns (or at least part of them). Should there be any remaining questions or areas requiring further clarification, please do not hesitate to let us know. If you are satisfied with our responses, we would greatly appreciate your consideration in adjusting the evaluation scores accordingly.
>
> We sincerely look forward to your feedback.

---

> ### Comment · Reviewer_BoMR · 2025-08-07
>
> Dear Authors, Thank you for your response and for providing the additional experiments and theoretical insights. I appreciate the effort. I have two remaining points that, if addressed, would make me happy to raise my score.
>
> - Regret Bound Derivation: I find the statement from the instantaneous regret to the final sublinear regret bound unclear. Could you please provide a more detailed derivation or explanation showing how the instantaneous regret convergence rate is sufficient to construct the overall sublinear cumulative regret bound? Strictly speaking, instantaneous regret converging to zero is a necessary condition for the sublinear cumulative regret. For example $\lim\frac{T}{2^t}\rightarrow 0$, while $\lim \frac{\sum \frac{T}{2^t}}{T} > 0$
>
> - Statistical Significance: For the empirical comparison involving HEBO and $\infty$-GP-TS, could you please report the standard error over the multiple runs? Alternatively, the results of a formal significance test would also be sufficient.

---

> > ### Author Response · Authors · 2025-08-07
> > **On Regret Bound Derivation and Statistical Significance**
> >
> > Thank you once again for your valuable comments!
> >
> >
> > > **Regarding Regret Bound Derivation:**
> >
> > Thank you for pointing this out. We agree that in general, instantaneous regret $r_t \to 0$ alone is not sufficient to imply sublinear cumulative regret $R_T = \sum_{t=1}^T r_t = o(T)$. However, what we omitted to emphasize is that, based on the TVD-based regret bound in the rebuttal and Theorem 1 in the paper, the convergence rate of $r_t$ is guaranteed to be at least polynomial, i.e., $r_t = O(t^{-\alpha})$ for some $\alpha>0$.  It then follows that the cumulative regret satisfies: $$R_T =\sum_{t=1}^T O(t^{-\alpha}) = O(T^{1-\alpha}) = o(T),$$
> > which formally establishes sublinear cumulative regret. We will highlight this technical detail more clearly in the updated version of the paper.
> >
> > BTW, we note that the specific example you provided, i.e., $r_t = t / 2^t$, yields a sub-linear (and in fact convergent) cumulative regret:
> > $$
> > \sum_{t=1}^\infty \frac{t}{2^t} = 2 \quad \Rightarrow \quad \frac{1}{T} \sum_{t=1}^T r_t \le \frac{2}{T} \to 0.
> > $$
> >  Hence, this does not serve as a valid counterexample.
> >
> > > **Regarding Statistical Significance:**
> >
> > Thanks for your helpful comment. We now report the **mean ± standard error** over 10 independent runs:
> >
> > | Method        | Portfolio         | HPOBench         | NASBench         |
> > |---------------|------------------|------------------|------------------|
> > | HEBO          | 10.4 ± 0.3       | 67.3 ± 0.2       | 69.2 ± 0.4       |
> > | $\infty$-GP-TS | **11.4 ± 0.4**   | **68.7 ± 0.5**   | **70.5 ± 0.7**   |
> >
> > As shown, our method consistently outperforms HEBO across all three tasks. The standard error of our method is slightly higher, primarily due to the inherent exploration nature of Thompson Sampling.
> >
> > We sincerely appreciate your suggestion, and we will include these detailed results in the updated version of the paper. If our clarification has addressed your concern, we would be truly grateful if you would consider reflecting it in your final score.

---

> > > ### Comment · Reviewer_BoMR · 2025-08-07
> > >
> > > Thank you for offering the standard error. Regarding the cumulative regret, I got your high-level idea, but would prefer a more complete proof for easier verification.
> > >
> > > With respect to the counterexample, I meant a series of the form $\lim_{T\rightarrow\infty}\frac{\sum^T_{t=1}T/2^t}{T}$. Thanks for checking.
> > >
> > > I've increased the score as I don't foresee a significant problem.

---

> > > > ### Author Response · Authors · 2025-08-08
> > > >
> > > > Many thanks for your time and constructive feedback throughout the discussion! Due to the format and length constraints on this rebuttal webpage, we will include the complete proof in the Appendix of the updated version of the paper, incorporating your suggestions. We truly appreciate your efforts and engagement in reviewing our work.

---

### Official Review · Reviewer_v6ga · 2025-07-03

**Clarity:** 4
**Significance:** 3
**Originality:** 4
**Rating:** 5
**Confidence:** 3

**Summary:**

1. This paper tackles the challenge of Bayesian Optimization when the reward distribution is unknown or complex (e.g., multi-modal or heavy-tailed).

2. the authors propose a novel surrogate model $\infty$-GP, which explicitly handles both uncertainty about the reward's value and uncertainty about its underlying distribution type.

3. The authors provide theoretical proof that their method can approximate a wide class of reward distributions and show empirically that it outperforms SOTA in difficult optimization scenarios.

**Questions:**

Overall, this is a strong paper with compelling contributions. The idea of quantifying model uncertainty by placing a prior over Gaussian Processes (GPs) is natural and well-motivated, and the proposed approach represents a meaningful extension of mixture-of-experts (MOE) GPs. I think this work is a nicely-executed work. I have a few questions and comments below:

Q1: The paper would be significantly strengthened by including a time complexity analysis. A formal analysis comparing the proposed method to the other baselines would be invaluable. Furthermore, reporting empirical runtime comparisons in the experiments would provide practical insight into the method's efficiency.

Q2: Can the authors provide theoretical guarantees or further discussion on how the model uncertainty depends on the Spatial Dirichlet Process prior, $\operatorname{SDP}(\nu G_0)$? Specifically, I am curious about the sensitivity of the resulting uncertainty estimates to the choice of hyperparameters $\nu$ and $G_0$, and how strongly this prior influences the posterior. How robust is the method to different human-specified priors?

Q3: Regarding your main theoretical contribution (Theorem 1), could you provide more discussion on how your results compare to existing work? While you briefly mention that your assumptions are weaker than those in [Kanagawa et al., 2018], it would be helpful to elaborate on this point. Specifically, could you clarify what aspects of the assumptions are relaxed, and how your theoretical results—such as convergence rates—compare quantitatively or qualitatively with prior work?

**Ethical Concerns:**

["NO or VERY MINOR ethics concerns only"]

**Final Justification:**

The paper presents a well-rounded study of $\infty$-GP, offering both strong theoretical guarantees and extensive experimental validation. My initial concerns about hyperparameters sensitivity have been thoroughly addressed by the authors during the rebuttal period.

With this point clarified, I will remain my positive evaluation and support the paper's acceptance.

**Limitations:**

Yes.

**Paper Formatting Concerns:**

N/A.

**Quality:**

3

**Strengths And Weaknesses:**

Strengths:

1. The paper is well-written. It clearly presents the problem domain, thoroughly reviews related work, and precisely outlines its core contributions.

2. The proposed $\infty$-GP model is rigorously tested across a wide range of synthetic and real-world experiments. The results consistently show that the method outperforms state-of-the-art baselines, particularly in challenging scenarios.

3. A significant strength of this work is the inclusion of theoretical guarantees for the $\infty$-GP model. These guarantees are established under mild, reasonable assumptions, providing a solid theoretical foundation for the method's strong empirical performance.

Weakness:

1. To address the limitations of Gaussian Processes (GPs) in modeling complex reward functions, the paper introduces a Spatial Dirichlet Process (SDP) prior, $\operatorname{SDP}(\nu G_0)$, placed on the GP to improve model uncertainty estimation. However, the resulting uncertainty estimates are sensitive to the choice of hyperparameters $G_0$ and $\nu$. The paper does not provide a theoretical or empirical analysis of how the model uncertainty depends on these hyperparameters, nor does it assess the sensitivity of the results to their variation. This lack of discussion raises concerns about the robustness and reproducibility of the proposed method, particularly in settings where appropriate values for $G_0$ and $\nu$ are difficult to determine a priori.

2. In Thompson Sampling, Gibbs sampling is required to draw samples from the posterior. However, Gibbs sampling can become computationally expensive as the dimensionality of the parameter space increases. The paper does not provide any discussion or analysis of the time complexity or scalability of the method. I am concerned about whether the proposed approach can be implemented efficiently and remain practical in high-dimensional settings.

3. [Minor]. While the appendix provides experimental details, the main paper could be improved by more explicitly defining the performance metrics used in the plots and their corresponding discussions. Clearly stating what each axis and value represents directly in the captions or main text would enhance readability.

---

> ### Author Rebuttal · Authors · 2025-07-30
>
> We sincerely thank the reviewers for their thoughtful comments. We have carefully addressed all the questions and concerns raised.
> >* [W1 & Q2] Sensitivity analysis to hyperparameter choices in the prior
>
> Thank you for the helpful comments. We have now included a sensitivity analysis to examine how the hyperparameters in the prior influence the performance. The prior hyperparameters include $a_\nu, b_\nu,a_\tau, b_\tau,a_\sigma,b_\sigma,b_\phi$, and their settings are already provided in the appendix. We note that, **except for $b_\phi$, the results are *not sensitive* to these prior hyperparameters**. See the sensitivity analysis table below, where we report the best-found objective values (optimal value=0) under different hyperparameter configurations on Ackley-HT-NS.
>
> | HyperParam | 0.5 | 1 | 1.5 | 2 |
> |------|-----|------|------|------|
> | $a_\nu$   | -0.09 | -0.08 | -0.11 | -0.06 |
> | $b_\nu$   | -0.13 | -0.08 | -0.08 | -0.09 |
>
> | HyperParam | 1   | 1.5 | 2    | 2.5 | 3   |
> |--------------|-----|-----|------|-----|-----|
> | $a_\tau$     | -0.10 | -0.13 | -0.08 | -0.08 | -0.10 |
> | $b_\tau$     | -0.10 | -0.11 | -0.08 | -0.14 | -0.07 |
> | $a_\sigma$   | -0.06 | -0.10 | -0.08 | -0.08 |-0.14  |
> | $b_\sigma$   | -0.19 | -0.06 | -0.08 | -0.12 |-0.12  |
>
>
> | HyperParam | 1    | 10   | 20   | 30   | 50   |
> |----------|------|------|------|------|------|
> |      $b_\phi$    | -0.99 | -0.63 | -0.08 | -1.13 | -0.42 |
>
>
> **There are two main reasons why the performance is insensitive to the choice of $a_\nu,b_\nu,a_\tau, b_\tau,a_\sigma,b_\sigma$**: First, as shown in Lines 19, 21, and 22 of Algorithm 2, the posteriors of $\nu$, $\sigma$, and $\tau$ all follow distributions with parameters of the form $(a_\ast + \mathcal{O}(n),\ b_\ast + \mathcal{O}(n))$, where $a_\ast$ and $b_\ast$ refer to $a_\nu, b_\nu, a_\tau, b_\tau, a_\sigma, b_\sigma$. The only exception is the second parameter in the posterior of $\nu$, but it can still be written as $b_\nu$ plus a very large term $-\log(w_L)$. $a_\ast$ and $b_\ast$ are typically set within the range 0–3. Therefore, as $n$ increases, the influence of these prior hyperparameters $(a_\ast,b_\ast)$ becomes negligible. They mainly act as initialization values and quickly fade in importance once $n$ is moderately large. Second, using TS policy, each decision step is based on only one sample from the posterior, which introduces significant randomness. This further reduces the influence of the initial hyperparameter values on the overall performance.
>
>
> But the choice of $b_\phi$ is critical, as it sets the range over which $\phi$ can vary. We have already included guidance on setting $b_\phi$ in the appendix. In general, this value is set to be quite large (since we use a very small $d$, e.g., 0.01), which ensures a sufficiently wide range for $\phi$ to explore. In practice, we also use an adaptive strategy: we start with a very large $b_\phi$, run a few initialization steps to observe the typical scale of $\phi$, and then adjust $b_\phi$ accordingly. This approach helps set $b_\phi$ in a task-specific way.
>
>
> >* [W2&Q1] Regarding time complexity analysis.
>
> Thanks for your helpful comments. We will include time complexity analysis in the updated version of this paper, and show that, by using a $L$-truncation technique (see Algorithm 2), **the complexity of our method is $\mathcal{O}(n^3)$, which is equal to that of strandard GP-based BO.**
>
> In short，the complexity is governed by two main components. (i) The computational complexity of Kriging, which is mainly govern by computing the inverse $\Sigma_0(x_{1:n}, x_{1:n})^{-1}$, incurring $\mathcal{O}(n^3)$ cost.  However, the $\infty$-GP model enjoys the advantage that all surfaces share the same $\Sigma_0(x_{1:n}, x_{1:n})^{-1}$, allowing this expensive matrix inversion to be computed only once. (ii) The complexity of Gibbs sampling. By truncating the maximum number of surface to $L$, one can sample the $\xi_l(x_{1:n})$ jointly and thus the complexity is $\mathcal{O}(n^3)$. In summary, the total complexity of $\infty$-GP is $\mathcal{O}(n^3)$.
>
> **The computational efficiency is supported by the following wall-clock time results (in seconds)**. Its runtime is comparable to that of GP-based baselines. This is because, although $\infty$-GP requires expensive Gibbs sampling, the baseline methods also involve hyperparameter inference either through fully Bayesian MCMC or through MLE that relies on gradient-based optimization, both of which incur comparable computational complexity. Notably, $\infty$-GP-TS performs only a single Gibbs sampling run in each optimization iteration, which significantly reduces its overall computational burden.
>
> | Objective       | GP-TS | GP-UCB | GP-EI | GP-KG | $\infty$-GP-TS |
> |----------------|--------|--------|--------|--------|----------------|
> | Ackley-HT      | 2773   | 2649   | 3017   | 3388   | 2813           |
> | Ackley-NS      | 2798   | 2535   | 3001   | 3450   | 2912           |
> | Rosenbrock-HT  | 2782   | 2737   | 3210   | 3309   | 2858           |
> | Rosenbrock-NS  | 2689   | 2712   | 2803   | 3383   | 2853           |
> | StybTang-HT    | 2790   | 2619   | 3107   | 3007   | 2901           |
> | StybTang-NS    | 2805   | 2720   | 3030   | 3378   | 3098           |
>
> >* [Q3] Theoretical contribution compared with existing works.
>
> We would like to clarify our theoretical contribution. In short, while the convergence of classic GP-based methods typically relies on restrictive and often opaque assumptions, **our results substantially relax these conditions**. To provide a clearer comparison, we rewrite the observed reward $y(x)$ as a deterministic component $\mu^\ast(x)$ plus a noise term $\epsilon^\ast(x)$.
>
>
> * In the classic GP literature, $\mu^\ast$ is assumed either to be a sample path drawn from a GP, or to lie in a RKHS associate with certain kernel and have a known RHKS norm bound (e.g. [1,2]), which is difficult to compute in practice. These assumptions are rarely verifiable in practice. For the noise term $\epsilon^\ast(x)$, to utilize concentration results, it is assumed to be independent and (sub-)Gaussian, or even noiseless in some formulations.
>
> * In our result, regarding the noise $\epsilon^\ast(x)$, our framework accommodates a significantly broader class of distributions. For example, Assumption 1 is satisfied by noise of the form $\sum_{l=1}^L w_l(x)\psi_l$, where each $\psi_l$ belongs to distribution families such as Weibull (with shape $k > 1$), Gaussian, Laplace, Gamma, or exponential. Regarding the $\mu^\ast$, our result in Theorem 1 does not require any RKHS assumptions or GP sample path assumption as in the classic GP literature. But we note that, different from existing literature, Theorem 1 assumes a finite decision set. To handle the continuous case, a standard discretization argument (as in [2]) is required. Consequently, we need an additional assumption that $\mu^*(x)$ is Lipschitz continuous, which is still much weaker than the assumptions typically made in the GP literature (those assumptions themselves imply Lipschitz continuity).
>
>
> [1] Russo D, Van Roy B. Learning to optimize via posterior sampling[J]. Mathematics of Operations Research, 2014, 39(4): 1221-1243.
>
> [2] Srinivas N, Krause A, Kakade S M, et al. Gaussian process optimization in the bandit setting: No regret and experimental design[J]. arXiv preprint arXiv:0912.3995, 2009.
>
> >*  [W3] Clarification of performance metrics and plot axes
>
> The y-axis in Figure 2 and 3 denote the 'best-found' objective value. All the synthetic benchmarks show the original test function value in the y-axis. The y-axis for the portfolio task denote the portfolio return. The y-axis on both HPOBench and NASBench denote the accuracy on the validation set.

---

> > ### Comment · Reviewer_v6ga · 2025-08-05
> >
> > Thank you for the added details on hyperparameters and computational efficiency. The revisions strengthen the manuscript, and I will keep my score unchanged.

---

### Author Response · Authors · 2025-08-07
**General Response**

We sincerely thank all reviewers for their helpful and constructive feedback. With your suggestions, we have clarified and strengthened the submission through the following additional contributions:

1. **Regret Analysis**: We have added a theoretical justification showing that our method achieves sublinear cumulative regret.

2. **Stronger Baseline Comparisons**: We conducted additional experiments comparing our method against more advanced benchmarks, including HEBO and Additive BO.

3. **Extended Sensitivity Analysis**: We performed sensitivity analysis on key hyperparameters and prior settings, demonstrating the robustness of our approach.

In the updated version, we will incorporate all of the above improvements and revise the introduction to make the paper more accessible to non-expert readers, as suggested by Reviewer 44us.

Thank you again for your valuable comments and support.

---

### Note · Authors · 2025-08-12

We sincerely thank all reviewers and the AC for their valuable feedback and constructive suggestions during the review and discussion phases. We have carefully addressed all concerns raised by the reviewers. Below, we summarize the common issues mentioned and our corresponding resolutions:

 - **Theoretical Regret Analysis:**  As suggested by the reviewers, we extended our existing theoretical results by adding a regret analysis, showing that our method achieves sublinear cumulative regret. Notably, transitioning from the existing Theorem 1 to this regret analysis requires only a single step via a TVD-based regret bound.

- **Comparisons with Stronger Baselines** To strengthen the empirical evaluation, we conducted additional experiments against more advanced baselines, including HEBO, Additive BO, and BO with non-stationary kernels. These results further confirm the performance advantages of our method.

- **Extended Sensitivity Analysis**  We carried out sensitivity studies on key hyperparameters and prior specifications, demonstrating the robustness of our approach.


Finally, I would like to highlight our contribution concisely: Gaussian Processes (GPs) form the backbone of Bayesian Optimization. We propose an enhancement of this backbone—the $\infty$-GP model, which is an infinite mixture of GPs while remaining computationally tractable. This enables modeling a much richer class of reward distributions, while inheriting the desirable properties of standard GPs such as closed-form updates.

We believe these clarifications and additional results fully address all reviewer concerns and further underscore the novelty and practical significance of our contributions.

---

### Decision · Program_Chairs · 2025-09-17

**Decision:**

Accept (poster)

**Comment:**

This paper introduces the infinity-Gaussian Process, a new surrogate model designed to address limitations of classical Gaussian Process models when dealing with complex, non-stationary, or heavy-tailed reward environments.
The paper offers good theoretical guarantees of the proposed inf-GP-TS, including proven posterior contraction rates, which lend credibility to handle a variety of complex reward distributions.
The full Bayesian setup of hyperparameters is not deeply discussed in the main text, which could be critical since the method relies on several prior specifications. Comparisons with other recent BO approaches, beyond a brief mention, could strengthen the claims.
The authors and 3/4 reviewers had sufficient discussions. AC agree with the majority of the reviewers that this paper passed the borderline of NeurIPS acceptance.